# Dynamics of the accelerated t-SNE

**Kyoichi Iwasaki**                                                                    *iwasaki.kyoichi@ism.ac.jp*
*The Graduate University for Advanced Studies, SOKENDAI*

**Hideitsu Hino**                                                                      *hino@ism.ac.jp*
*The Institute of Statistical Mathematics*

**Reviewed on OpenReview:** *https://openreview.net/forum?id=4205*

## Abstract

This paper investigates the dynamics of t-Stochastic Neighbor Embedding (t-SNE), a popular tool for visualizing complex datasets in exploratory data analysis, optimized by the Nesterov's accelerated gradient method. Building on the foundational work that connects t-SNE with spectral clustering and dynamical systems, we extend the analysis to include accelerated dynamics which is not addressed in the previous work, revealing the emergence of Bessel and modified Bessel functions as a novel aspect of the algorithm's behavior characterizing the temporal evolution of the accelerated t-SNE. Because the ordinary differential equation corresponding to the optimization process under consideration has a closed-form solution, by performing eigenvalue decomposition of the data's adjacency matrix as a preprocessing step, we can obtain low-dimensional embeddings at any point in time without performing sequential optimization. This advancement not only enhances the practical utility of t-SNE but also contributes to a deeper understanding of its underlying dynamics.

## 1 Introduction

Data visualization plays a pivotal role in exploratory data analysis, enabling the comprehension of complex datasets by reducing them to more cognitively manageable two or three-dimensional representations. Among various dimensionality reduction techniques, t-distributed stochastic neighbor embedding(t-SNE) (van der Maaten & Hinton, 2008), a descendant of the stochastic neighbor embedding (Hinton & Roweis, 2002), has emerged as powerful tools for visualizing the overall structure of data and understanding how clusters are formed across diverse fields. t-SNE continues to undergo technical and theoretical development, and its applications are also expanding. In (Zhu & Ting, 2022), the use of an isolation kernel instead of a Gaussian kernel accelerates computation, while (Böhm et al., 2023) integrates contrastive learning within computer vision research, and (Skrodzki et al., 2023) explores the concept of coarse embedding to expedite inference. Furthermore, the concept has been extended into hyperbolic space using polar quadtree techniques, as demonstrated in related research by (Zhou & Sharpee, 2021; Guo et al., 2022). Also, theoretical advancements have been made, such as the convergence study presented in (Jeong & Wu, 2024). For general low-dimensional representation, the concept of dynamical dimension reduction is introduced in (Yoon & Osting, 2022).

In this paper, we focus on another aspect of t-SNE, the acceleration of the optimization procedure from the viewpoint of dynamical system. In many software implementation of t-SNE (Szymański & Kajdanowicz, 2017; Krijthe, 2015), the optimization procedure includes a momentum term. We consider applying momentum-based methods such as the Momentum Method (MM) and the Nesterov's accelerated gradient (NAG) method (Nesterov, 1983) to t-SNE aiming to enhance the convergence behavior of the optimization dynamics in the early exaggeration (EE) stage. We analyze the impact of these modifications from a continuous-time dynamical systems perspective.

We note that the t-SNE is composed of two stages: the EE stage and the *embedding* stage (van der Maaten & Hinton, 2008). Some previous theoretical studies (Linderman & Steinerberger, 2019; Cai & Ma, 2022) have

conducted analyses focusing on the EE stage. While these methods are often introduced to accelerate practical optimization, our primary focus here is on understanding the dynamical behavior of the system rather than achieving computational speed-up. Since this stage itself possesses a sufficiently rich mathematical structure, we will also focus on the EE stage, and the embedding stage is not modified in this study.

## 1.1 Contributions

We draw upon the foundational work of (Cai & Ma, 2022), who established a connection between t-SNE clustering dynamics and spectral clustering under certain conditions for gradient descent (GD) method, framing the data clustering phenomenon within a dynamical systems perspective. While the use of MM or NAG can potentially accelerate the optimization in the EE stage, it is not obvious whether the resulting embeddings still preserve cluster structures in the same way as standard GD. Our contribution extends this analysis by demonstrating that the incorporation of MM and NAG can accelerate the EE dynamics; in particular, we show that NAG introduces mathematical entities such as the Bessel and modified Bessel functions into the system. Through rigorous theoretical analysis and empirical validation, we show that our approach offers a further insights into the dynamical properties of t-SNE.

Our main contributions are summarized as follows:

- We explore an approach of dynamical system analysis of t-SNE including a momentum term, and further extended to the NAG. We find that in the former case (MM), even with the same linear ODE, convergence is theoretically faster [Proposition 5.4], and in the case of the NAG, it leads to the derivation of related ODEs as the first kind Bessel functions or modified Bessel functions. [Theorem 6.4] As these claims suggest, our approach explicitly involves eigenvalues and eigenvectors. When the data size is $n$, solving the eigenvalue problem requires computational complexity $O(n^3)$, which is greater than the original $O(n^2)$ complexity. Therefore, in this paper, we limit our numerical experiments to datasets of relatively small volume.

- We derive these ODEs and their relations, organizing them into several theorems, propositions and lemmas, where all proofs are provided in the Appendix. As a matter of fact, deriving parallel results already given without momentum in (Cai & Ma, 2022) to those with momentum term are technically not straight forward.

- Since the closed-form solution of the ODE is available, embeddings at any point in time can be easily obtained. [Proposition 5.4, Theorem 6.4]

- We find that insights related to EE stage in (Cai & Ma, 2022) are replicated in our method with acceleration, leading to the conclusion that the optimization algorithm converges in less time than when using GD. Since the EE stage does not have an explicitly defined objective function with the acceleration parameter $\alpha > 1$, it is challenging to monitor the function value and determine an appropriate stopping point during optimization. As a secondary benefit of our continuous formulation via ODEs and their closed-form solutions, we can define a principled stopping criterion based on a threshold applied to the Accumulated Residual Ratio (ARR) of major and minor eigenvalue contributions. [(27), (28)] This approach enables the identification of a suitable termination time within the continuous framework. Such a mechanism is particularly advantageous when t-SNE is employed as part of a larger system or pipeline, where an automatic and interpretable stopping condition is beneficial.

- We theoretically demonstrate that, under certain conditions, iterative computations for both MM and NAG can be replaced by their continuous counterparts by adopting this continuous relaxation approach using ODEs. [Proposition5.2, 6.2] It provides theoretical justification for results that had previously been obtained heuristically through iterative methods. We can confidently rely on the original discrete optimization algorithms (MM and NAG), as they retain the essential dynamics while being more computationally efficient. However, since solving the ODEs typically requires eigenvalue decompositions of large matrices, the practical advantage of using the continuous formulation is limited.

- By interpreting the spectral decomposition of the dynamics in terms of Lyapunov exponents, we offer a novel theoretical perspective on the formation of clusters in t-SNE. [PropositionH.1, H.2] Components associated with positive Lyapunov exponents dominate the long-term behavior and effectively determine the clustering structure, with the corresponding eigenvectors encoding cluster assignments. [(29)] In contrast, components with negative Lyapunov exponents decay over time and gradually vanish from the embedding. Notably, in the case of NAG, these stable components do not simply decay exponentially but exhibit non-monotonic, oscillatory behavior governed by Bessel functions.

## 1.2 Key take aways for t-SNE users

We investigate both the theoretical and practical aspects of MM and NAG, based on the framework of GD. Our study builds upon the theoretical foundation established by (Cai & Ma, 2022), which demonstrated that the EE stage of the t-SNE algorithm can be approximated by a first-order ODE. The solution of this ODE can be expressed explicitly in terms of the adjacency matrix's eigenvalues and eigenvectors, and providing a theoretical justification for why t-SNE reveals the data's cluster structure.

In our paper, we extend this line of reasoning by investigating whether a similar theoretical framework can be applied to other optimization schemes such as MM and NAG. Specifically, we ask: Can the dynamics of MM and NAG during the EE stage also be described using ODEs, and if so, do the same assurances hold?

The answer to this question is "yes": we theoretically demonstrate that optimizing t-SNE with either MM or NAG likewise yields the emergence of clear cluster structures. This result implies that, although the computational cost of optimization was previously a concern for t-SNE, one can now employ accelerated gradient methods with confidence. Furthermore, our work proposes a novel stopping criterion for the ODE-based formulation, a point which is not explicitly addressed in (Cai & Ma, 2022)'s original work. This contributes both to the theoretical understanding and practical usability of accelerated variants of t-SNE.

Next, we present the practitioner-oriented takeaway. We derive the corresponding ODEs and their closed-form solutions by continuously relaxing the step-by-step iterative process. This relaxation enables, particularly in the EE stage, the approximation of intermediate states at arbitrary time points by evaluating the ODE solution. In addition, we propose a stopping criterion, referred to as the ARR metric defined in (28), which provides a principled method for determining an appropriate termination time. To apply our approach in practice, two major assumptions must be taken into account:

- When the data size $n$ is sufficiently small, the eigenvalue problem required by the method—typically of computational complexity $O(n^3)$—can be solved within a practical amount of time.

- The assumptions stated in the theoretical takeaway—namely, (I1), (T1.M), (T2.M) for the MM method, and (I1), (T1.N), (T2.N) for NAG—are satisfied.

Provided these conditions hold, the following approach becomes applicable:

- Preparation: Compute the adjacency matrix $\mathbf{P}$ and its eigenvalues.

- EE stage: Compute the ARR-based criterion and determine the stopping time.

- Embedding stage: Obtain the low-dimensional embedding by evaluating the formula (3) with $m^{(k+1)} = 0$, where acceleration is unnecessary in order to obtain stable results in the embedding stage.

## 2 Related work

Dealing with t-SNE, a method of dimensionality reduction, as a dynamical system requires a background in both the methodology of dimensionality reduction and analysis of algorithms as dynamical systems. Moreover, the unique properties of t-SNE and the development of related research are also pertinent. We concisely summarize the related studies for each aspect.

## 2.1 Dimensional reduction methodology

In dimensionality reduction, numerous methods have been established to transform high-dimensional data into a more manageable, lower-dimensional representation, aiding in elucidation of the data structure. Principal component analysis (PCA) (Jolliffe, 1986) serves as a foundational linear method, identifying principal axes of maximum variance. Isometric mapping (Isomap) (Tenenbaum et al., 2000) focuses on preserving the geometric distances on the manifold, effectively maintaining the spatial relationships among data points. Locally linear embedding (LLE) (Roweis & Saul, 2000) isolates local linear relationships by reconstructing points from their immediate neighbors, reflecting the intrinsic geometry of the manifold. The Laplacian eigenmap technique (Belkin & Niyogi, 2001) constructs a graph Laplacian and leverages its eigenfunctions to achieve a lower-dimensional projection that reveals the manifold's local characteristics.

By further developing graph-based methods such as Isomap and LLE, SNE (Hinton & Roweis, 2002) and t-SNE (van der Maaten & Hinton, 2008) have been proposed as methods that learn embeddings directly and have become very popular. UMAP (McInnes & Healy, 2018) and PHATE (Moon et al., 2019) have been developed as powerful alternatives to the t-SNE algorithm, and recently, the relationships between t-SNE and UMAP have also been discussed (Damrich et al., 2023).

## 2.2 Gradient methods to differential equation

The approach of deriving and analyzing ODEs as continuous limits of iterative optimization algorithms is widely adopted. This includes attempts to extend acceleration techniques beyond NAG (Wibisono et al., 2016), employing Runge-Kutta integrators (Zhang et al., 2018) and symplectic integrators (Goto & Hino, 2025), and formulating continuous-time models using Lyapunov analysis and tools from stochastic calculus (Orvieto & Lucchi, 2019). These studies exemplify the breadth of research exploring different aspects of continuous approaches to optimization. There are more geometrically sophisticated methods, such as (Defazio, 2019), which addresses NAG on a Riemannian manifold, and (Wilson et al., 2021), which performs analysis using Lyapunov functions. In deriving the ODEs related to the NAG method, we refer to several results from (Su et al., 2016), where the emergence of Bessel functions for a related ODE is established.

## 2.3 SNE algorithm and recent developments

In addition to the studies introduced in the introduction, variants of t-SNE and relationships with other methods are also being actively researched (Kobak et al., 2019; Chen et al., 2015; Pezzotti et al., 2017; Fujiwara et al., 2020). A generic formulation of embedding algorithms that includes SNE and other existing algorithms are studied, elucidating their relation with spectral methods and graph Laplacians (Carreira-Perpiñán, 2010; Vladymyrov & Carreira-Perpiñán, 2012).

There are several theoretical attempts, such as (Linderman & Steinerberger, 2019), which compares spectral clustering and t-SNE and (Arora et al., 2018), which states that t-SNE works well under specific conditions called $\gamma$-spherical and $\gamma$-separated in EE stage. The paper (Linderman & Steinerberger, 2022) presents theoretical and experimental results, focusing on sufficient conditions for the parameters $\alpha$ for acceleration, and $h$ for step size under the assumption of GD. It also discusses considerations regarding disjoint clusters and the independence of the results from initial values. Note that such results include both EE and subsequent embedding stages.

Additionally, recent studies have addressed aspects related to implicit regularization and scaling effects in clustering algorithms. For instance, (Auffinger & Fletcher, 2023) discusses implicit regularization mechanisms, as highlighted in Proposition 6.5, which connects closely to early stopping in clustering optimization processes. Additionally, (Murray & Pickarski, 2024) explores the scaling effects arising from the gradual decrease of $p_{ij}$, which is the element of adjacency matrix commonly used in t-SNE is the symmetric probability matrix $\mathbf{P} = (p_{ij})$, as $n \to \infty$. This phenomenon appears to relate to Lemma 5.5 or 6.3, particularly the condition concerning the sign of $\alpha \lambda_R(\mathbf{L}(\mathbf{P})) - \frac{1}{n-1}$, where $\lambda_R(\mathbf{L}(\mathbf{P}))$ stands for $R^{th}$-eigenvalue of the unnormalized Laplacian matrix of $\mathbf{P}$. These findings provide critical insights into how regularization and scaling behaviors influence clustering performance and stability under large-scale settings.

### 2.4 Acceleration methods

It is widely known that a drawback of t-SNE is its high computational cost, and many attempts have been made to improve its computational efficiency. The original paper (van der Maaten & Hinton, 2008) uses both early exaggeration effect by adopting acceleration parameter $\alpha > 1$ and momentum methods. The Barnes-Hut tree is a well-known approach for accelerating the t-SNE optimization procedure (van der Maaten, 2013; 2014). The n-body force calculations is utilized in (Yang et al., 2013; Vladymyrov & Carreira-Perpiñán, 2014). (Skrodzki et al., 2024) applies Barnes-Hut approximation idea into hyperbolic disc to accelerate low-dimensional embedding. We note that it is experimentally shown in (Lambert et al., 2023) that NAG can be an alternative to EE.

Although the term "acceleration" often suggests computational speed-up, we clarify that our current framework focuses on theoretical acceleration in the continuous-time dynamics, rather than practical runtime efficiency. In fact, due to the reliance on full eigendecomposition of the graph Laplacian, the overall computational complexity remains $O(n^3)$ in our current implementation. Nonetheless, we believe that this spectral formulation offers valuable analytical insight, and that future implementations using approximate eigensolvers (e.g., Lanczos algorithm) may help bridge the gap between theoretical structure and practical scalability.

For example, the oscillatory behavior revealed through Bessel-type dynamics is directly captured in the ARR metric we propose, which provides an interpretable and data-driven criterion for early stopping. This analytical formulation could inspire new techniques for adaptive exaggeration scheduling or spectrally informed initialization, potentially improving both the interpretability and efficiency of t-SNE variants in large-scale applications.

## 3 Original t-SNE algorithm

We explain the original t-SNE algorithm (van der Maaten & Hinton, 2008). For a vector $\mathbf{a} = (a_i)_{1 \leq i \leq n} \in \mathbb{R}^n$, define its $l^p$ norm by $\|\mathbf{a}\|_p = (\sum_{i=1}^n |a_i|^p)^{1/p}$. For a matrix $\mathbf{A} = (a_{ij}) \in \mathbb{R}^{n \times n}$ and its spectral norm is $\|\mathbf{A}\| = \sup_{\|\mathbf{x}\|_2 \leq 1} \|\mathbf{A}\mathbf{x}\|_2$. We also introduce sets of orthogonal matrices $O(n,k) = \{\mathbf{V} \in \mathbb{R}^{n \times k}; \mathbf{V}^\top \mathbf{V} = \mathbf{I}_k\}$ and $O(n) = O(n,n)$. We use the notation $[n] = \{1, 2, \ldots, n\}$ for $n \in \mathbb{Z}_{>0}$. The diameter of a set $S(\subseteq \mathbb{R}^n)$ is $\text{diam}(S) = \sup_{x,y \in S} \|x - y\|_2$. For a symmetric matrix $\mathbf{A} = (a_{ij})_{1 \leq i,j \leq n} \in \mathbb{R}^{n \times n}$, we define degree operator $\mathbf{D} : \mathbb{R}^{n \times n} \to \mathbb{R}^{n \times n}$ as $\mathbf{D}(\mathbf{A}) = \text{diag}(\sum_{i=1}^n a_{i1}, \ldots, \sum_{i=1}^n a_{in})$, and Laplacian operator $\mathbf{L} : \mathbb{R}^{n \times n} \to \mathbb{R}^{n \times n}$ by $\mathbf{L}(\mathbf{A}) = \mathbf{D}(\mathbf{A}) - \mathbf{A}$. Also, we set $\mathbf{1}_n = (1, \ldots, 1)^\top \in \mathbb{R}^n$ and $\mathbf{H}_n = \frac{1}{n(n-1)}(\mathbf{1}_n \mathbf{1}_n^\top - \mathbf{I}_n) \in \mathbb{R}^{n \times n}$.

Let $\{X_i\}_{i \in [n]} \subseteq \mathbb{R}^{d_h}$ be the $d_h$-dimensional data points. For the pairs of data points $\{(X_i, X_j)\}_{1 \leq i \neq j \leq n}$, we define a symmetric matrix $\mathbf{P} = (p_{ij})_{1 \leq i,j \leq n} \in \mathbb{R}^{n \times n}$ with $p_{ii} = 0$ for $\forall i \in [n]$ and $p_{ij} = (p_{i|j} + p_{j|i})/(2n)$ for $i \neq j$, where

$$p_{j|i} = \frac{\exp(-\|X_i - X_j\|_2^2/2\tau_i^2)}{\sum_{l \in \{1,2,\ldots,n\} \setminus \{i\}} \exp(-\|X_i - X_l\|_2^2/2\tau_i^2)}. \tag{1}$$

Here, $\tau_i > 0$ is the bandwidth parameter. The aim of t-SNE is to achieve a low-dimensional representation $\{y_i\}_{i \in [n]} \subseteq \mathbb{R}^{d_l}$, $d_l < d_h$ of $\{X_i\}_{i \in [n]}$. Typically, $d_l$ is two or three because t-SNE is predominantly used for visualization, and in this paper, we consider $d_l = 2$ without loss of generality. Consider a symmetric matrix $\mathbf{Q} = (q_{ij})_{1 \leq i,j \leq n}$ where $q_{ii} = 0$ for $\forall i \in [n]$ and

$$q_{ij} = \frac{(1 + \|y_i - y_j\|_2^2)^{-1}}{\sum_{l,s \in \{1,2,\ldots,n\}, l \neq s} (1 + \|y_l - y_s\|_2^2)^{-1}} \tag{2}$$

for $i \neq j$. We get the low dimensional representation $\{y_i\}_{i \in [n]}$ by minimizing the Kullback-Leibler (KL) divergence (Kullback & Leibler, 1951) between matrices $\mathbf{P}$ and $\mathbf{Q}$ as $(y_1, \ldots, y_n) = \arg\min_{y_1,\ldots,y_n} \sum_{i,j \in [n], i \neq j} p_{ij} \log \frac{p_{ij}}{q_{ij}}$.

As described in (Cai & Ma, 2022), the t-SNE algorithm is typically implemented in two distinct phases: an EE stage followed by an embedding stage. While previous studies have explored accelerating the EE stage by introducing a parameter $\alpha > 1$ into the GD framework, this paper extends the concept further

by investigating whether analogous acceleration strategies can be systematically applied to MM and NAG. Although the explicit form of the objective function for the case with $\alpha > 1$ is not known, we proceed by assuming the following formulation in line with the original t-SNE work (van der Maaten & Hinton, 2008) and (Cai & Ma, 2022). The following update rule aligns to (Cai & Ma, 2022), which states

$$y_i^{(k+1)} = y_i^{(k)} + h \sum_{1 \leq i,j \leq n, \ i \neq j} (y_j^{(k)} - y_i^{(k)}) S_{ij}^{(k)}(\alpha) + m^{(k+1)}(y_i^{(k)} - y_i^{(k-1)}), \tag{3}$$

where $h \in \mathbb{R}^+$ represents the step size parameter, originally denoted as $4h$ in (van der Maaten & Hinton, 2008), and $m^{(k+1)} \in \mathbb{R}$ is a momentum parameter. The value $\alpha > 0$, known as the exaggeration parameter, plays a crucial role in the gradient coefficient $S_{ij}^{(k)}(\alpha)$ derived in Appendix A, and is given by

$$S_{ij}^{(k)}(\alpha) = \frac{\alpha p_{ij} - q_{ij}^{(k)}}{1 + \|y_i^{(k)} - y_j^{(k)}\|_2^2} \in \mathbb{R}, \tag{4}$$

where $\sum_{1 \leq i,j \leq n, i \neq j} (y_j^{(k)} - y_i^{(k)}) S_{ij}^{(k)}(\alpha) \in \mathbb{R}^2$, and we define adjacency matrix $\mathbf{S}_\alpha^{(k)} = (S_{ij}^{(k)}(\alpha))_{i,j \in [n]}$. The algorithm starts with an initialization $y_i^{(0)} = y_i^{(-1)}$, $\forall i \in [n]$. Although the momentum term $m^{(k+1)}(y_i^{(k)} - y_i^{(k-1)})$ to accelerate the updating algorithm was not discussed for simplicity in (Cai & Ma, 2022), our aim is to discuss how we may accelerate the convergence of the algorithm and derive a dynamical system corresponding to accelerated updating methods.

## 4 Asymptotic behavior of the update equation

The points $\{y_i^{(k)}\}_{i \in [n]} \subseteq \mathbb{R}^2$ are represented in two dimensions, but they are reorganized into $n$ vectors according to the first and second coordinates as $\mathbf{y}_l^{(k)} \in \mathbb{R}^n, l \in [2]$. This identification and the following Theorem 4.1 enable us to make sure that the adjacency matrices $\mathbf{S}_\alpha^{(k)}$ is considered as a fixed matrix $\alpha \mathbf{P} - \mathbf{H}_n$. For $l \in [2]$, Eq. (3) is reformulated as

$$\mathbf{y}_l^{(k+1)} = [\mathbf{I}_n - h\mathbf{L}(\mathbf{S}_\alpha^{(k)})]\mathbf{y}_l^{(k)} + m^{(k+1)}(\mathbf{y}_l^{(k)} - \mathbf{y}_l^{(k-1)}), \tag{5}$$

where $\mathbf{I}_n \in \mathbb{R}^{n \times n}$ is the identity matrix and $\mathbf{y}_l^{(k)} \in \mathbb{R}^n$ is the $l$-th coordinates of $\{y_i^{(k)}\}_{i \in [n]}$. The following theorem shows that the matrix $\mathbf{S}_\alpha^{(k)} = (S_{ij}^{(k)}(\alpha))_{i,j \in [n]}$ can be approximately simplified. This approximation, combined with a continuous relaxation with $H_n$, enables the derivation of ODEs from the recurrence relation (3), making it one of the core statements of this paper.

**Theorem 4.1.** *(Asymptotic Graphical interpretation, Theorem 2 in (Cai & Ma, 2022)) Let $\mathbf{P} = (p_{ij})_{1 \leq i,j \leq n}$ be a symmetric matrix defined in Eq. (1) and denote $\eta^{(k)} := (\mathrm{diam}(\{y_i^{(k)}\}_{1 \leq i \leq n}))^2$. Then, for any $i, j \in [n]$ with $i \neq j$, and each $k \geq 1$ such that $\eta^{(k)} < 1$, we have*

$$\left| S_{ij}^{(k)}(\alpha) - \alpha p_{ij} + \frac{1}{n(n-1)} \right| \leq \alpha p_{ij} \eta^{(k)} + \frac{2\eta^{(k)}}{n(n-1)(1 - \eta^{(k)})}. \tag{6}$$

*Then for each $k \geq 1$, as long as $\eta^{(k)}$ and $\alpha$ satisfy $\eta^{(k)} \ll \frac{\|\mathbf{P}\|}{n\|\mathbf{P}\|_\infty}$, $\alpha \gg \frac{1}{n\|\mathbf{P}\|}$, we have*

$$\lim_{n \to \infty} \frac{\|\mathbf{S}_\alpha^{(k)} - (\alpha \mathbf{P} - \mathbf{H}_n)\|}{\|\alpha \mathbf{P} - \mathbf{H}_n\|} = 0. \tag{7}$$

With this statement, Eq. (5) admits an approximation

$$\mathbf{y}_l^{(k+1)} \approx [\mathbf{I}_n - h\mathbf{L}(\alpha \mathbf{P} - \mathbf{H}_n)]\mathbf{y}_l^{(k)} + m^{(k+1)}(\mathbf{y}_l^{(k)} - \mathbf{y}_l^{(k-1)}). \tag{8}$$

We consider the global behavior on $\{\mathbf{y}_l^{(k)}\}$, $l \in [2]$ under the following initialization and condition of parameters [1].

---

[1] The condition (I1) comes from 'Initialization', and (T1) describes 'Trajectory' as $n \to \infty$.

**(I1)** $\{y_i^{(0)}\}_{i\in[n]}$, $\{y_i^{(1)}\}_{i\in[n]}$ satisfy $\min_{l\in[2]}\{\|\mathbf{y}_l^{(0)}\|_2, \|\mathbf{y}_l^{(1)}\|_2\} > 0$ and
$\max_{l\in[2]}\{\|\mathbf{y}_l^{(0)}\|_\infty, \|\mathbf{y}_l^{(1)}\|_\infty\} = O(1)$ as $n \to \infty$.

**(T1)** The parameters $(\alpha, h, k)$ satisfy $k(nh\alpha\|\mathbf{P}\|_\infty + h/n) = O(1)$ as $n \to \infty$.

Condition (I1) states that the initialization $\{y_i^{(0)}\}$ should not be simply all zeros or unbounded, and (T1) says that the cumulative differences of $h\mathbf{L}(\mathbf{S}_\alpha^{(k)})$ and $h\mathbf{L}(\alpha\mathbf{P} - \mathbf{H}_n)$ are limited.

The update formula (8) is a three-term recurrence relation and two initial vectors $\mathbf{y}_l^{(0)}$ and $\mathbf{y}_l^{(1)}$ need to be set. With the momentum term, the vectors $\mathbf{y}_i^{(k+1)}$ stays within limited area:

**Proposition 4.2.** *(Localization) Under the condition (I1) and (T1), we have*

$$\text{diam}(\{y_i^{(k+1)}\}_{i\in[n]}) \leq C \max_{l\in[2]}\{\|\mathbf{y}_l^{(0)}\|_\infty, \|\mathbf{y}_l^{(1)}\|_\infty\} \tag{9}$$

*for some universal constant $C > 0$.*

The proposition 4.2 ensures that the data points $\{\mathbf{y}_i^{(k)}\}$ are globally bounded and the condition (I1) leads to $\eta^{(k)} < 1$, and both the theorem 4.1 and the approximate Eq. (8) holds. This result had been derived in (Cai & Ma, 2022) without the momentum term; however, deriving this result when considering the momentum term is not trivial and requires an evaluation of the difference in solutions at two consecutive points $t-1$ and $t$ originating from the momentum term.

## 5 Gradient flow with constant momentum coefficient

We assume that $m^{(k+1)} = m \in (0, 1)$, which is commonly assumed in existing works (Kovachki & Stuart, 2021; He et al., 2023; Hao et al., 2021; Sutskever et al., 2013; Rashidi et al., 2020). Let $\{\tilde{\mathbf{y}}_l^{(k)}\}_{k\geq 0}$ be computed by the update equation

$$\tilde{\mathbf{y}}_l^{(k+1)} = [\mathbf{I}_n - h\mathbf{L}(\alpha\mathbf{P} - \mathbf{H}_n)]\tilde{\mathbf{y}}_l^{(k)} + m(\tilde{\mathbf{y}}_l^{(k)} - \tilde{\mathbf{y}}_l^{(k-1)}). \tag{10}$$

The following lemma derives the ODE that holds for MM in the continuous-time limit as $h \to 0$.

**Lemma 5.1.** *With the assumption $\tilde{\mathbf{y}}_l^{(k)} \approx \mathbf{Y}_l(kh)$, $t = kh$ and the continuous limit $h \to 0$, the update rule (10) is reduced to the ODE with initial value $\tilde{\mathbf{y}}_l^{(0)} = \mathbf{Y}_l(0)$*

$$\dot{\mathbf{Y}}_l(t) = -\frac{1}{1-m}\mathbf{L}(\alpha\mathbf{P} - \mathbf{H}_n)\mathbf{Y}_l(t). \tag{11}$$

Furthermore, the following proposition provides a theoretical guarantee that $\tilde{\mathbf{y}}_l^{(k)}$ and $\mathbf{Y}_l(kh)$ become sufficiently close as $h \to 0$.

**Proposition 5.2.** *(Gradient flow with constant momentum) Under the condition (I1), for $l \in [2]$, $k \in \mathbb{Z}_{\geq 0}$, and some positive constants $C_1, C_2$, we have*

$$\frac{\|\tilde{\mathbf{y}}_l^{(k)} - \mathbf{Y}_l(kh)\|_2}{\|\mathbf{Y}_l(0)\|_2} \leq \frac{2khmC_1}{1-m}\|\mathbf{L}(\alpha\mathbf{P} - \mathbf{H_n})\| + \frac{k}{2}\Big(\frac{h}{1-m}\Big)^2\|\mathbf{L}(\alpha\mathbf{P} - \mathbf{H_n})\|^2 C_2. \tag{12}$$

This proposition justifies that the approximation $\tilde{\mathbf{y}}_l^{(k)} \approx \mathbf{Y}_l(kh)$ works under the conditions (I1), (T1). To better understand the solution of the derived ODE (11), we consider its eigendecomposition. Remember that the matrix $\mathbf{P}$ is symmetric, and its Laplacian $\mathbf{L}(\mathbf{P})$ is likewise. With the eigendecomposition of $\mathbf{L}(\mathbf{P})$ expressed as $\mathbf{L}(\mathbf{P}) = \mathbf{V}^\top \mathbf{\Lambda} \mathbf{V}$, we have

**Lemma 5.3.** *The matrix $\mathbf{L}(\alpha\mathbf{P} - \mathbf{H}_n)$ is symmetric, and we have eigendecomposition:*

$$\mathbf{L}(\alpha\mathbf{P} - \mathbf{H}_n) = \mathbf{V}^\top \Sigma \mathbf{V}, \tag{13}$$

*where $\Sigma = \mathrm{diag}(\sigma_i)_{i \in [n]}$, $\mathbf{\Lambda} = \mathrm{diag}(\lambda_i)_{i \in [n]}$, $0 = \lambda_1 \leq \lambda_2 \leq \ldots \leq \lambda_n$ are eigenvalues[2] and $\mathbf{V} \in O(n)$, which are the same eigenvectors of $\mathbf{L}(\mathbf{P})$. Here, the relation between $\Sigma$ and $\mathbf{\Lambda}$ is*

$$\sigma_1 = \lambda_1 = 0, \sigma_i = \alpha\lambda_i - \frac{1}{n-1} \ (2 \leq i \leq n). \tag{14}$$

With the Lemma 5.1, we achieve the explicit expression

$$\mathbf{Y}_l(t) = \exp\Big(-\frac{t}{1-m}\mathbf{L}(\alpha\mathbf{P} - \mathbf{H}_n)\Big)\mathbf{y}_l^{(0)}. \tag{15}$$

Combining the eigendecomposition and Eq. (15) yields the following proposition:

**Proposition 5.4.** *(Solution path on constant momentum) For $l \in [2]$, the first order linear ODE (11) with initial value $\mathbf{Y}_l(0) = \mathbf{y}_l^{(0)}$ and momentum term $m \in (0,1)$ has the unique solution:*

$$\mathbf{Y}_l(t) = (\mathbf{u}_1^\top \mathbf{y}_l^{(0)})\mathbf{u}_1 + \sum_{i=2}^{n} \exp\left(-\frac{t}{1-m}\left(\alpha\lambda_i - \frac{1}{n-1}\right)\right)(\mathbf{u}_i^\top \mathbf{y}_l^{(0)})\mathbf{u}_i. \tag{16}$$

Propositions 5.2 and 5.4 extend (21) or Proposition 8 in (Cai & Ma, 2022) with a constant momentum term, as substituting $m = 0$ in Eq. (16) recovers the original result. The effect of momentum coefficients is discussed in Appendix I.7.

## 5.1 Well-conditioned matrix

The behavior of $\exp\left(-\frac{t}{1-m}\left(\alpha\lambda_i - \frac{1}{n-1}\right)\right)$ for $t \gg 1$ depends on the sign of $\alpha\lambda_i - \frac{1}{n-1}$. If negative, the corresponding components are amplified; if positive, they are suppressed. It implies that

$$\lim_{t \to \infty} \mathbf{Y}_l(t) \in \mathrm{span}(\{\mathbf{u}_1, \ldots, \mathbf{u}_R\}). \tag{17}$$

If the data $\{X_i\}_{i \in [n]}$ are well clustered and bandwidths $\tau_i$ are properly chosen, (Balakrishnan et al., 2011) demonstrates that $\mathbf{P}$ is well approximated. We assume the weighted graph of a well-conditioned matrix $\mathbf{P}^*$, as detailed in Appendix E, has $R \geq 2$ connected components.

## 5.2 Spectral convergence with constant momentum coefficient

Condition(T1) ensures Theorem 4.1 holds. We assume a stronger, analogous condition (T1.M), along with (T2.M), which governs eigenvalues and clustering behavior in section 5.1:

**(T1.M)** The parameters $(\alpha, h, t)$ satisfy $\alpha \gg [n\lambda_{R+1}(\mathbf{L}(\mathbf{P}))]^{-1}$ and $\frac{t}{1-m} = o(n)$ as $n \to \infty$.

**(T2.M)** There exists a symmetric and well-conditioned matrix $\mathbf{P}^* \in \mathbb{R}^n$ such that $\lambda_{R+1}(\mathbf{L}(\mathbf{P}^*)) \gg \max\{(\frac{t\alpha}{1-m})^{-1}, \|\mathbf{L}(\mathbf{P}^* - \mathbf{P})\|\}$ and $\frac{t\alpha}{1-m}\|\mathbf{L}(\mathbf{P}^* - \mathbf{P})\| = o(1)$ as $n \to \infty$.

In the context of visualization and clustering, the following lemma is fundamental in identifying which components remain dominant in the eigendecomposition representation.

**Lemma 5.5.** *Under the conditions (T1.M), (T2.M) and $n \gg 1$, we have*

$$\alpha\lambda_R(\mathbf{L}(\mathbf{P})) - \frac{1}{n-1} \leq 0, \quad \alpha\lambda_{R+1}(\mathbf{L}(\mathbf{P})) - \frac{1}{n-1} > 0. \tag{18}$$

*Especially, the eigenvalues $\{\sigma_i\} \subseteq \mathbb{R}$ consist of $R$-negative numbers or zeros ($0 = \sigma_1, \sigma_2, \ldots, \sigma_R$), and $n - R$ positive numbers ($\sigma_{R+1}, \ldots, \sigma_n$).*

---

[2]We denote $i$-th eigenvalue as $\lambda_i(\mathbf{A})$, if we emphasize the original matrix $\mathbf{A}$.

Note that this proposition is not explicitly stated in (Cai & Ma, 2022), but it implies that (18) holds under the conditions (T1.M) and (T2.M). This, in turn, means that the same $R$ eigenvalues and eigenvectors as in the GD case with $m = 0$ remain in the visualization, thereby providing a theoretical guarantee that MM serves as an acceleration of GD.

# 6 Gradient flow with NAG

Another accelerated gradient method is NAG (Nesterov, 1983), which takes the following form: starting with $\tilde{\mathbf{y}}_l^{(0)}$ and $\mathbf{w}_l^{(0)} = \tilde{\mathbf{y}}_l^{(0)}$ under our situation:

$$\tilde{\mathbf{y}}_l^{(k+1)} = \mathbf{w}_l^{(k)} - h\mathbf{L}(\alpha\mathbf{P} - \mathbf{H}_n)\mathbf{w}_l^{(k)}, \tag{19}$$

$$\mathbf{w}_l^{(k)} = \mathbf{y}_l^{(k)} + \frac{k-1}{k+2}(\tilde{\mathbf{y}}_l^{(k)} - \tilde{\mathbf{y}}_l^{(k-1)}). \tag{20}$$

Similar to Lemma 5.1, we have another linear ODE with NAG.

**Lemma 6.1.** *With the assumption $\tilde{\mathbf{y}}_l^{(k)} \approx \mathbf{Y}_l(k\sqrt{h})$ for a smooth curve $\mathbf{Y}_l(t)$ for $t \geq 0$ and $l \in [2]$, and putting $k = t/\sqrt{h}$, we have the ODE corresponding to the update rule of NAG in Eqs. (19), (20) as*

$$\ddot{\mathbf{Y}}_l(t) + \frac{3}{t}\dot{\mathbf{Y}}_l(t) + \mathbf{L}(\alpha\mathbf{P} - \mathbf{H}_n)\mathbf{Y}_l(t) = 0. \tag{21}$$

The following proposition provides a theoretical guarantee that $\tilde{\mathbf{y}}_l^{(k)}$ and $\mathbf{Y}_l(k\sqrt{h})$ holds for NAG, just as Proposition 5.2.

**Proposition 6.2.** *As the step size $h \to 0$, the update rule (19), (20) is reduced to the ODE (21) in the sense that for all fixed $T > 0$,*

$$\lim_{h \to 0} \max_{0 \leq k \leq \frac{T}{\sqrt{h}}} \|\tilde{\mathbf{y}}_l^{(k)} - \mathbf{Y}_l(k\sqrt{h})\|_2 = 0. \tag{22}$$

We aim to demonstrate that this situation would be reasonable under the following conditions:

**(T1.N)** The parameters $(\alpha, h, t)$ satisfy $\alpha \gg [n\lambda_{R+1}(\mathbf{L}(\mathbf{P}))]^{-1}$ and $t = o(n^{\frac{1}{2}})$ as $n \to \infty$.

**(T2.N)** There is a symmetric and well-conditioned matrix $\mathbf{P}^* \in \mathbb{R}^n$ such that $\lambda_{R+1}(\mathbf{L}(\mathbf{P}^*)) \gg \max\{(t^2\alpha)^{-1}, \|\mathbf{L}(\mathbf{P}^* - \mathbf{P})\|\}$, and $t^2\|\mathbf{L}(\mathbf{P}^* - \mathbf{P})\| = o(1)$ as $n \to \infty$.

Similar to Lemma 5.5, we have the following lemma:

**Lemma 6.3.** *Under the conditions (T1.N), (T2.N) and $n \gg 1$, we have*

$$\alpha\lambda_R(\mathbf{L}(\mathbf{P})) - \frac{1}{n-1} \leq 0, \quad \alpha\lambda_{R+1}(\mathbf{L}(\mathbf{P})) - \frac{1}{n-1} > 0. \tag{23}$$

*Especially, the entire eigenvalues $(\sigma_1, \ldots, \sigma_n)$ consist of $R$-negative numbers or zeros $(0 = \sigma_1, \sigma_2, \ldots, \sigma_R)$, and $n - R$ positive numbers $(\sigma_{R+1}, \ldots, \sigma_n)$.*

In the case of NAG, this proposition guarantees that, under the conditions (T1.N) and (T2.N), the same $R$ eigenvalues and eigenvectors are emphasized in visualization and clustering, and that the number $R$ matches that in both GD and MM as Lemma 5.5.

As in Lemma 5.3, we have another closed-form expression by solving linear ODE (21).

**Theorem 6.4.** *Under the conditions (T1.N), (T2.N), partition the eigenvectors as $\mathbf{V} = (\mathbf{U}, \mathbf{U}_\perp)$, where $\mathbf{U} \in O(n, R)$, $\mathbf{U}_\perp \in O(n, n - R)$* [3]. *Then, we have*

$$\mathbf{Y}_l(t) = \mathbf{U}\mathbf{\Gamma}_1(t)\mathbf{y}_{l,1:R}^{(0)} + \mathbf{U}_\perp\mathbf{\Gamma}_2(t)\mathbf{y}_{l,R+1:n}^{(0)}, \tag{24}$$

---

[3] $\mathbf{U}_\perp$ is orthogonal complement of $\mathbf{U}$.

*where*

$$\mathbf{\Gamma}_1(t) = \mathrm{diag}\left(1, \frac{2}{t\sqrt{-\sigma_2}}I_1(t\sqrt{-\sigma_2}), \ldots, \frac{2}{t\sqrt{-\sigma_R}}I_1(t\sqrt{-\sigma_R})\right), \mathbf{\Gamma}_2(t) = \mathrm{diag}\left(\frac{2}{t\sqrt{\sigma_{R+1}}}J_1(t\sqrt{\sigma_{R+1}}), \ldots, \frac{2}{t\sqrt{\sigma_n}}J_1(t\sqrt{\sigma_n})\right),$$

*and the symbol $I_1(\cdot)$ denotes the modified Bessel function of the first kind, while $J_1(\cdot)$ represents the Bessel function of the first kind. Also, $\mathbf{y}_{l,1:R}^{(0)} = (\mathbf{y}_{l,1}^{(0)}, \ldots, \mathbf{y}_{l,R}^{(0)})^\top \in \mathbb{R}^R$, $\mathbf{y}_{l,R+1:n}^{(0)} = (\mathbf{y}_{l,R+1}^{(0)}, \ldots, \mathbf{y}_{l,n}^{(0)})^\top \in \mathbb{R}^{n-R}$ are the initial values.*

The explicit expression (24) is the solution path under the NAG for the t-SNE algorithm. In the Appendix I.1, we consider the qualitative differences in the solutions of ODEs corresponding to GD, MM, and NAG by plotting the solutions.

Lemmas 5.5 and 6.3 demonstrate that eigenvalues and eigenvectors for which $\alpha\lambda_i - \frac{1}{n-1} \leq 0$ are emphasized, highlighting key features for spectral clustering. This phenomenon, where large eigenvalues $\lambda_i$ decrease with increasing $t$, is referred to as implicit regularization. The subsequent proposition asserts that $\mathbf{Y}_l(t)$ converges towards the subspaces spanned by the first $R$ eigenvectors, which form the Laplacian null space commonly utilized in spectral clustering, thus embodying implicit regularization with the inclusion of momentum terms. Furthermore, the assumptions regarding $t$ such as $t = o(n)$, $t = o(n^{1/2})$, $\frac{t\alpha}{1-m}\|\mathbf{L}(\mathbf{P}^* - \mathbf{P})\| = o(1)$ or $t^2\alpha\|\mathbf{L}(\mathbf{P}^* - \mathbf{P})\| = o(1)$ imply necessity for early stopping to avoid overshooting.

**Proposition 6.5.** *(Implicit regularization, clustering and early stopping) Under the conditions (I1), (T1.M) and (T2.M), let $\mathbf{Y}_l(t)$ be the function defined as (16), $\mathbf{U}_0 \in O(n, R)$ be such that the columns span the null space of $\mathbf{P}^*$. Then we have*

$$\lim_{n\to\infty} \frac{\|\mathbf{Y}_l(t) - \mathbf{U}_0\mathbf{U}_0^\top\mathbf{Y}_l(t)\|_2}{\|\mathbf{Y}_l(0)\|_2} = 0, \quad l \in [2]. \tag{25}$$

*Also, for a permutation matrix $O \in \mathbb{R}^{n\times n}$, we have*

$$\lim_{n\to\infty} \frac{\|\mathbf{Y}_l(t) - O\mathbf{z}_l\|_2}{\|\mathbf{Y}_l(0)\|_2} = 0, \quad l \in [2], \tag{26}$$

*where $\mathbf{z}_l = (z_{l1}, \ldots, z_{l1}, z_{l2}, \ldots, z_{l2}, \ldots, z_{lR}, \ldots, z_{lR})^\top \in \mathbb{R}^n$ and $z_{lr} = \boldsymbol{\theta}_r^\top y_l^{(0)}/\sqrt{n_r}$ for $r \in [R]$, the number of $z_{lr}$ is $n_r$, i.e. the number of nodes in the $r$-th connected component. Similarly, under conditions (I1), (T1.N) and (T2.N), let $\mathbf{Y}_l(t)$ be the function defined in (24) $\mathbf{U}_0 \in O(n, R)$ be that the columns span the null space of $\mathbf{P}^*$. Then we have the same formulas (25), (26).*

This proposition shows that in both cases, $\mathbf{Y}_l(t)$ serves as eigenvectors defining the Laplacian null space, producing results equivalent to spectral clustering. Despite utilizing the same data points and the same adjacency matrix $\mathbf{P}$ with corresponding eigenvalues and eigenvectors, the increase in $t$ presupposes distinct formulations: the MM adheres to $t = o(n)$ and $t\alpha\|\mathbf{L}(\mathbf{P}^* - \mathbf{P})\| = o(1)$ as $n \to \infty$, whereas the NAG relies on $t = o(n^{1/2})$ and $t^2\alpha\|\mathbf{L}(\mathbf{P}^* - \mathbf{P})\| = o(1)$ as $n \to \infty$. It is consistent with the general observation that the NAG converges more rapidly than the MM. Also, Lyapunov exponents are discussed in Appendix H.

## 7 Stop timing

While commonly used metrics such as Trustworthiness (Venna & Kaski, 2001) and Continuity (Kaski et al., 2003) are effective in evaluating the quality of low-dimensional embeddings after dimensionality reduction, they are not suitable for determining the evaluation timing in the context of ODEs. To address this limitation, we propose a novel metric, Average Residual Ratio(ARR). As will be confirmed in Section 8.2 and I.6, the comparison between the proposed ARR and the Trustworthiness values of the embedding indicates a well-aligned correspondence. The term 'stop time' is used informally to refer to a general criterion for deciding when to stop the algorithm, and should not be confused with the technical notion of stopping time in stochastic analysis. With Eqs. (16), (24), we express $\mathbf{Y}_l(t) = \sum_{i=1}^n c_{i,l}(t)\mathbf{u}_i$ with the time-variant coefficient $c_{i,l}(t) \in \mathbb{R}$. To evaluate the positive contribution of $c_{i,l}(t)$, define $a_i(t) = \sum_{l\in[2]} |c_{i,l}(t)|$. With Eq. (14) and in the case of MM, $a_i(t) = \sum_{l\in[2]} \exp\left(-\frac{t\sigma_i}{1-m}\right)|\mathbf{u}_i^\top\mathbf{y}_l^{(0)}|$, where we set $m = 0$ for GD. By contrast, in the

case of NAG,

$$a_i(t) = \begin{cases} \sum_{l \in [2]} \left| \frac{2\mathbf{y}_{l,i}^{(0)}}{t\sqrt{-\sigma_i}} I_1(t\sqrt{-\sigma_i}) \right|, & (i = 1, \ldots, R) \\ \sum_{l \in [2]} \left| \frac{2\mathbf{y}_{l,i}^{(0)}}{t\sqrt{\sigma_i}} J_1(t\sqrt{\sigma_i}) \right|, & (i = R+1, \ldots, n). \end{cases} \tag{27}$$

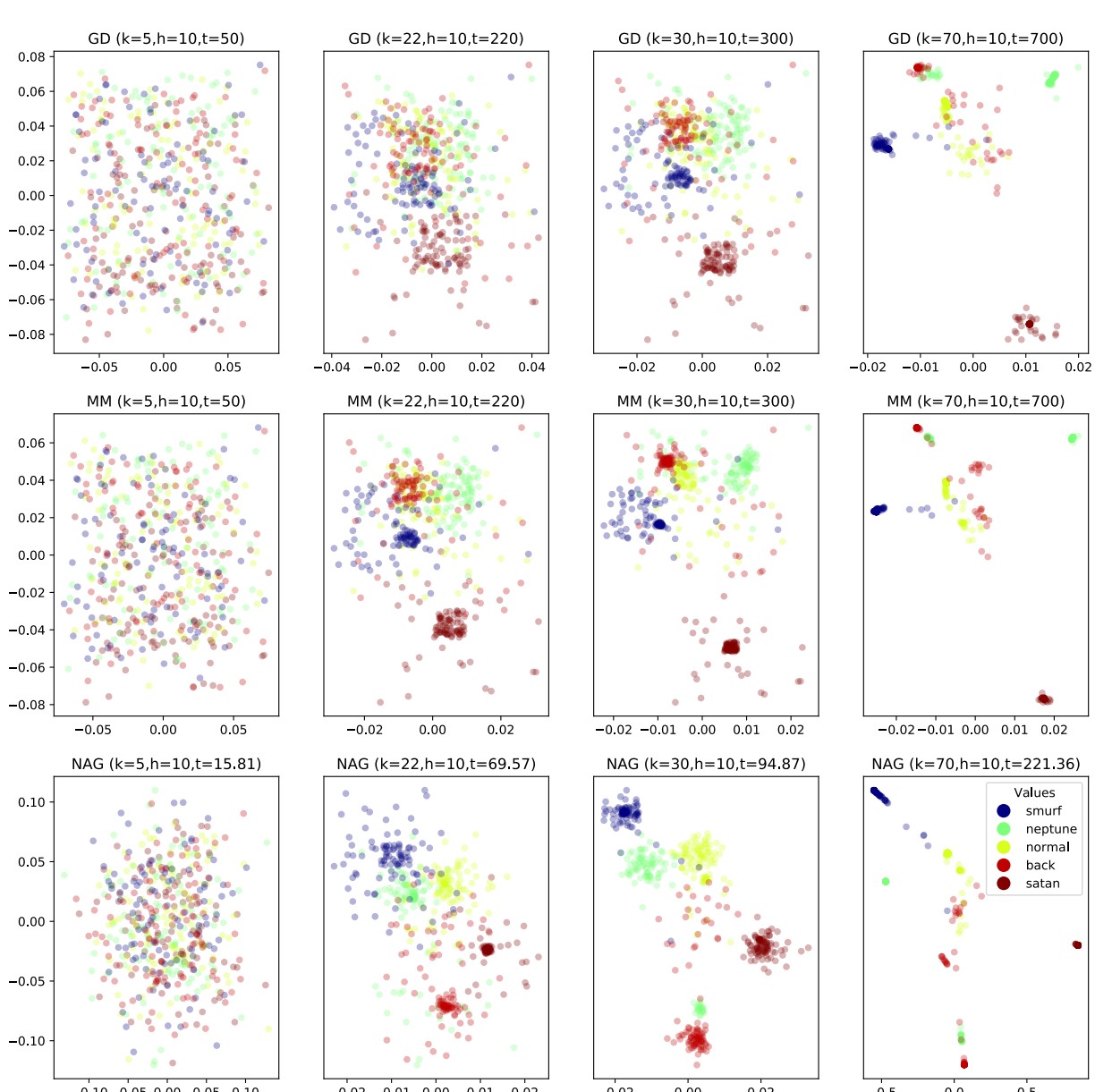

Figure 1: t-SNE visualization of $n = 500$ KDD Cup 1999 dataset using three ODEs: GD, MM and NAG.

To determine the stopping time, we propose the Average Residual Ratio(ARR), which divides the equation into main terms $(i = 1, .., R)$ and residuals $(i = R+1, .., n)$, based on the average contribution to clustering:

$$ARR(t) = \frac{\frac{1}{n-R} \sum_{i=R+1}^{n} a_i(t)}{\frac{1}{R} \sum_{i=1}^{R} a_i(t) + \frac{1}{n-R} \sum_{i=R+1}^{n} a_i(t)}. \tag{28}$$

Based on the Lemmas 5.5, 6.3 and the observation of Figures 4a, 4b in Appendix, it is expected that $a_i(t)$ increase monotonically for $i = 1, \ldots, R$, and decreasing or gradually decreasing for $i = R+1, \ldots, n$.

We propose a stopping criterion based on a threshold (e.g., 0.01) and demonstrate its effectiveness in subsection 8.2. While t-SNE is unsupervised, this heuristic is similar to those used for convergence in algorithms like GD/MM/NAG. Our analysis shows that components with negative eigenvalues form clusters, while those with positive eigenvalues don't contribute to the final embedding. Thus, using the ratio of these components as a convergence criterion for the ODE is reasonable.

### 7.1 Lyapunov exponents

The ARR used to determine the stop timing was based on the time parameter $t$, but considering the solution of the ODE as a dynamical system reveals an intriguing relationship.

The Lyapunov exponents for MM and NAG are given by H.122, H.123. Lemmas 5.5, 6.3 show that the first $R$ eigenvalues are non-positive. Let the Lyapunov exponent for MM be $\lambda_{Lyap,MM,i}$ and that for NAG be $\lambda_{Lyap,NAG,i}$. When momentum parameter $m$ does not get close to 1, it is observed that

$$\lambda_{Lyap,MM,i} = -\frac{1}{1-m}\Big(\alpha\lambda_i - \frac{1}{n-1}\Big) < \sqrt{-\alpha\lambda_i + \frac{1}{n-1}} = \lambda_{Lyap,NAG,i}, \text{for } i \in [R]. \tag{29}$$

This is guaranteed by the fact that the function of $x$ satisfies $0 < x < 1$ and $x < \sqrt{x}$, and a larger Lyapunov exponent implies faster cluster convergence. This is consistent with the commonly stated notion that NAG achieves faster convergence as an optimization method.

## 8 Numerical experiments

We numerically evaluate our results by comparing the t-SNE algorithms with and without acceleration and their ODE counterparts with real-world datasets: KDDcup1999 and MNIST[4]. Results on other data are shown in Appendix I. In this paper, we set perplexity = 30, as in Cai & Ma (2022) [5]. In all experiments in this section, all initial embedding vectors were generated randomly. The discussion on the methods of initialization in t-SNE and the impact of different initializations, along with experimental results, are presented and discussed in Appendices I.4 and I.5.

### 8.1 Experiment 1: Dynamic behavior of clustering over time

The KDDCup1999 dataset (Stolfo & Chan, 1999; Stolfo et al., 1998; Tavallaee et al., 2009) includes a wide variety of intrusions simulated in a military network environment, and comprises 42 columns with their labels. We extracted 100 samples of data for each of the 5 labels ('smurf', 'neptune', 'normal', 'back', 'satan') from the dataset, totaling 500 samples.

For the solution of ODEs corresponding to GD, MM and NAG, we substitute concrete value of the time $t$ to obtain two dimensional map of the original points in 42 dimensional space in Fig. 1, where the top, middle and bottom rows correspond to results of GD, MM and NAG, respectively. The column is aligned with the same iteration count k, and as we move to the right, $k$ increases, with the corresponding time $t$ increasing accordingly. The first row corresponds to GD, the second to MM, and the third to NAG. Comparing GD and MM, as described in Eq. (16), the effect of the momentum term $m$ causes the results to appear as though time $t$ has been fast-forwarded, even for the same time point. As for NAG, it can be observed that cluster formation occurs earlier in time $t$. Specifically, while the second column of MM corresponds to $t = 220$, the fourth column of NAG at $t = 221.36$ already shows that cluster formation is nearly complete. We state the assumption in Lemma 5.1 about GD and MM, that is we identify the variables as $t = kh$, where the variables $k$ is iteration number of the original iterative algorithms, $h$ as step size and $t$ is time variable. For NAG, the identification can be $t = k\sqrt{h}$ as in Lemma 6.1.

---

[4]The experiments were conducted on a laptop PC with a 12th Gen Intel(R) Core(TM) i7-1255U processor, 500GB storage, 16GB memory, using Python.

[5]Perplexity is typically set between 5 and 50, as mentioned in van der Maaten & Hinton (2008).

Let's pick up the first MM case ($t = 50$) and the second NAG case ($t = 69.57$). The variables $t$ are formally different, but for MM, due to the presence of the momentum term $m = 0.5$, it effectively becomes $t = 69.57 (= 50/(1 - 0.5))$ (See the formula in Proposition 5.4, where we adopt that MM can be understood as fast-forwarding GD with respect to the variable $t$.). Although they have practically similar variable $t$, the results are drastically different. The three plots in the rightmost column are all elongated.

Furthermore, in NAG, the final plot on the rightmost panel corresponds to $t = 221.36$, while in the MM, the second plot from the left corresponds to $t = 314.29$. The application conditions for the NAG are either $t = o(n^{1/2})$ in (T1.N) and $t^2 \|\mathbf{P}^* - \mathbf{P}\| = o(1)$ condition in (T2.N). By contrast, the application conditions for the MM are either $\frac{t}{1-m} = o(n)$ in (T1.M) and the condition $\frac{t}{1-m} \|\mathbf{P}^* - \mathbf{P}\| = o(1)$ in (T2.M). This generally suggests that the NAG can only be applied for relatively smaller values compared to the MM.

## 8.2 Experiment 2: Stopping criterion

Although we have conducted experiments to demonstrate that the accelerated method can indeed achieve embeddings in fewer time steps in the previous section, in our analysis, the embeddings are explicitly obtained as solutions to differential equations. Therefore, while this research originated from accelerating t-SNE, it no longer makes sense to compare it with other methods from the perspective of computational efficiency. We demonstrate that by setting a threshold to ARR (28), we obtain a reasonable embedding without using an iterative optimization algorithm with a popular MNIST dataset, which contains grayscale images of handwritten digits, and we focus on 400 images with 4 labels('2', '4', '6', '8') totaling 1600 samples, where each image contains $28 \times 28 = 784$ pixels. At the same time, we provide a comparison with standard Trustworthiness metric (Venna & Kaski, 2001), which is commonly used in dimensionality reduction methods such as t-SNE.

Figure 2 shows the transition of ARR (28) and Trustworthiness with 10 nearest neighbors over the time $t$ (left) and the embedding results for each method: GD, MM, and NAG after early exaggeration stage (right-top) and embedding stage (right-bottom).

As $t$ increases, ARR decreases, with NAG being the fastest, followed by MM, and GD being the slowest. The ARR crosses the 0.01 threshold at $t = 2700$ for GD, $t = 1400$ for MM, and $t = 280.0$ for NAG. The right panels show that the embeddings at these times provide reasonable clustering. As noted in section 7.1, cluster formation occurs fastest with NAG, followed by MM and GD. This can be interpreted by analyzing the Lyapunov exponents of the underlying ODE. When comparing ARR and Trustworthiness, as ARR decreases, Trustworthiness tends to increase, showing a general inverse correlation. The correlation coefficient was approximately $-0.77$. More details are reported in Appendix I.6.

As shown in the original paper (van der Maaten & Hinton, 2008) and the subsequent study in (Cai & Ma, 2022), the t-SNE algorithm typically visualizes clustering results by performing an embedding stage following the early exaggeration stage, which intend to facilitate cluster formation. The visualization in the lower right of Figure 2 follows this convention. It is important to note that the embedding stage is solely for visualization purposes. Therefore, no acceleration methods such as MM or NAG were applied. Instead, GD was iteratively used after the early exaggeration stage.

## 9 Conclusion

In this paper, we derive a linear ODE and Bessel differential equation corresponding to the MM and NAG in optimizing t-SNE with explicit solutions. In addition, akin to results obtained through GD, performing an eigenvalue decomposition confirms that implicit regularization, specifically clustering around small eigenvalues near zero, which is crucial in spectral clustering, similarly holds for them.

In addition to its significance as a theoretical analysis of the t-SNE algorithm—a popular method for dimensionality reduction and visualization—there is practical utility in that the ODE we derive has a closed-form solution. This means that by performing eigenvalue decomposition of the adjacency matrix in advance, we have the advantage of obtaining embedding results at any desired point in time.

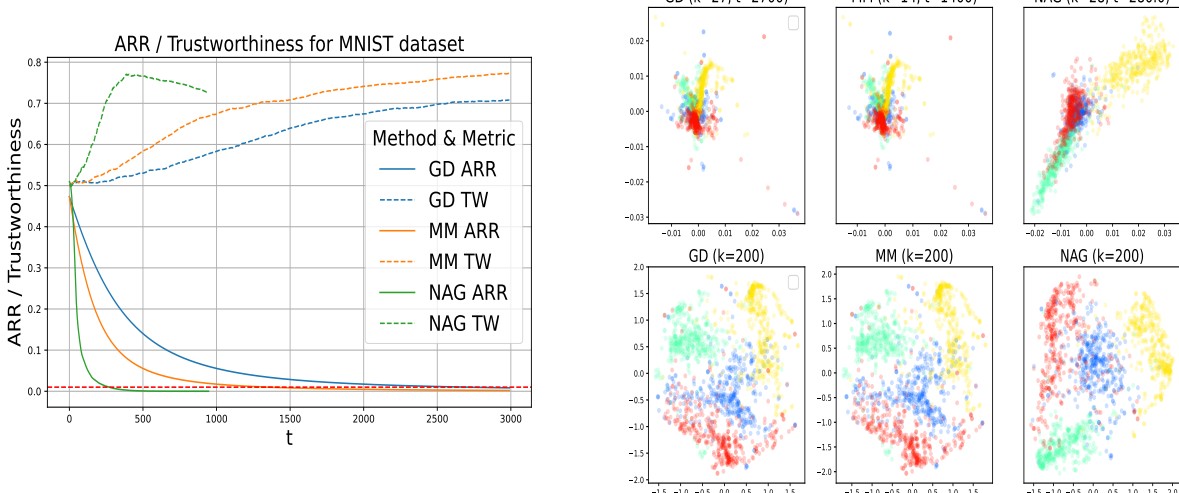

Figure 2: (left): transition of ARR and Trustworthiness (TW). (right): clustering result for MNIST dataset with ARR=0.01 using three ODEs GD, MM, and NAG (right-top), and result after the following embedding stage (right-bottom)

While this framework currently relies on a full eigendecomposition of the graph Laplacian, which incurs $O(n^3)$ complexity, we emphasize that this step is used primarily for theoretical analysis and computation of diagnostic metrics such as ARR. For large-scale applications, we anticipate that approximate methods such as the Lanczos algorithm or randomized eigendecomposition could reduce the computational burden to around $O(n^2)$, especially for sparse graphs. Exploring these approximations would be a promising direction for extending our framework to high-dimensional datasets at scale.

We discuss the approximated formula for t-SNE only concerning the EE stage but do not address the embedding stage. Theoretical analysis of the embedding stage will complement our work and further deepen the understanding of t-SNE. Additionally, we incorporate bandwidth $\tau_i$ into the perplexity but only examined specific case. A broader analysis of different scenarios and their implications on perplexity would provide a more comprehensive understanding.

UMAP (Ghojogh et al., 2021) has been frequently used alongside t-SNE. There are studies discussing the relationship between t-SNE and UMAP (Damrich et al., 2023); hence, analyzing UMAP through a dynamical systems-based approach is an intriguing direction for future research.

## Acknowledgement

Part of this work is supported by JSPS KAKENHI (23K24909 and 25H01494). Finally, we express our special thanks to the editor and anonymous reviewers whose valuable comments helped to improve the manuscript.

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

# A  Derivation of the t-SNE gradient coefficient

For the sake of brevity, we omit the iteration index $k$, we use $y_i$, $p_{ij}$, $q_{ij}$, instead of $y_i^{(k)}$, $p_{ij}^{(k)}$ and $q_{ij}^{(k)}$. The contents basically follows to Appendix A in van der Maaten & Hinton (2008).

t-SNE minimizes the KL divergence between the joint probabilities $p_{ij}$ in the high-dimensional space $\mathbb{R}^{d_h}$ and the joint probabilities $q_{ij}$ in the low-dimensional space $\mathbb{R}^{d_l}$. They are defined in Eq. (1), $p_{ij} = (p_{i|j} + p_{j|i})/2n$ and Eq. (2), respectively. Cost function $C$ is defined with KL divergence Kullback & Leibler (1951):

$$C = KL(P\|Q) = \sum_{ij} p_{ij} \log \frac{p_{ij}}{q_{ij}} = \sum_{ij} \left( p_{ij} \log p_{ij} - p_{ij} \log q_{ij} \right). \tag{A.1}$$

For the purpose of the derivation briefly, we set two variables $d_{ij}$ and $Z$:

$$d_{ij} = \|y_i - y_j\| \tag{A.2}$$

$$Z = \sum_{k \neq l} (1 + d_{kl}^2)^{-1}. \tag{A.3}$$

Then, with the chain rule, the gradient of the cost function $C$ with respect to $y_i$ is given by

$$\frac{\partial C}{\partial y_i} = \sum_j \left( \frac{\partial C}{\partial d_{ij}} + \frac{\partial C}{\partial d_{ij}} \right)(y_i - y_j)/d_{ij} \tag{A.4}$$

$$= 2\sum_j \frac{\partial C}{\partial d_{ij}}(y_i - y_j)/d_{ij}. \tag{A.5}$$

$p_{ij}$ are defined with high-dimensional data points, and they're irrelevant to $y_i$ or $d_{ij}$ that is $\frac{\partial p_{kl}}{\partial d_{ij}} = 0$ for $\forall k, l \in [n]$. Then, we have

$$\frac{\partial C}{\partial d_{ij}} = -\sum_{k \neq l} p_{kl} \frac{\partial \log q_{kl}}{\partial d_{ij}} \tag{A.6}$$

$$= -\sum_{k \neq l} p_{kl} \frac{\partial(\log q_{kl}Z - \log Z)}{\partial d_{ij}} \tag{A.7}$$

$$= -\sum_{k \neq l} p_{kl} \left( \frac{1}{q_{kl}Z} \frac{\partial((1 + d_{kl}^2)^{-1})}{\partial d_{ij}} - \frac{1}{Z} \frac{\partial Z}{\partial d_{ij}} \right). \tag{A.8}$$

The gradient term $\frac{\partial((1+d_{ij}^2)^{-1})}{\partial d_{ij}}$ is only nonzero when $k = i$ and $l = j$. Then,

$$\frac{\partial C}{\partial d_{ij}} = 2\frac{p_{ij}}{q_{ij}Z}(1 + d_{ij}^2)^{-2}d_{ij} - 2\sum_{k \neq l} p_{kl} \frac{(1 + d_{ij}^2)^{-2}d_{ij}}{Z}. \tag{A.9}$$

Note that $\sum_{k \neq l} p_{kl} = 1$ and we have

$$\frac{\partial C}{\partial d_{ij}} = 2p_{ij}(1 + d_{ij}^2)^{-1}d_{ij} - 2q_{ij}(1 + d_{ij}^2)^{-1}d_{ij} = 2(p_{ij} - q_{ij})(1 + d_{ij}^2)^{-1}d_{ij}. \tag{A.10}$$

Finally, we have

$$\frac{\partial C}{\partial y_i} = 4\sum_j \frac{p_{ij} - q_{ij}}{1 + \|y_i - y_j\|^2}(y_i - y_j). \tag{A.11}$$

In this formula, we aim to emphasize the effect of the original data distribution using $p_{ij}$. We formally substitute $p_{ij}$ with $\alpha p_{ij}$ and define the term $\frac{\alpha p_{ij} - q_{ij}}{1 + \|y_i - y_j\|^2}$ as $S_{ij}(\alpha)$. This leads to the expression in Eq. (4).

# B The effect of accelerator parameter

In this section, we examine the effect of the acceleration parameter $\alpha$ separately for GD and MM/NAG.

## B.1 The case of GD

First, consider the update equation (B.12) for GD. The low-dimensional representations $y_i^{(k)}$ and $y_j^{(k)}(i \neq j)$ show that $y_i^{(k+1)}$ is calculated using $y_i^{(k)}$ and a gradient term. The sign of $\alpha p_{ij} - q_{ij}^{(k)}$ determines its directional influence. We define $\alpha p_{ij}$ as the attractive force and $q_{ij}^{(k)}$ as the repulsive force. When $\alpha p_{ij} - q_{ij}^{(k)} > 0$, meaning the attractive force is stronger, $y_i^{(k+1)}$ moves closer to $y_j^{(k)}$ compared to $y_i^{(k)}$. Conversely, when $\alpha p_{ij} - q_{ij}^{(k)} < 0$, the repulsive force dominates, causing $y_i^{(k+1)}$ to move further away from $y_j^{(k)}$. Figure 3 illustrates this situation.

$$y_i^{(k+1)} = y_i^{(k)} + h \sum_{1 \leq i,j \leq n, \ i \neq j} \frac{\alpha p_{ij} - q_{ij}^{(k)}}{1 + \|y_i^{(k)} - y_j^{(k)}\|_2^2}(y_j^{(k)} - y_i^{(k)}) \tag{B.12}$$

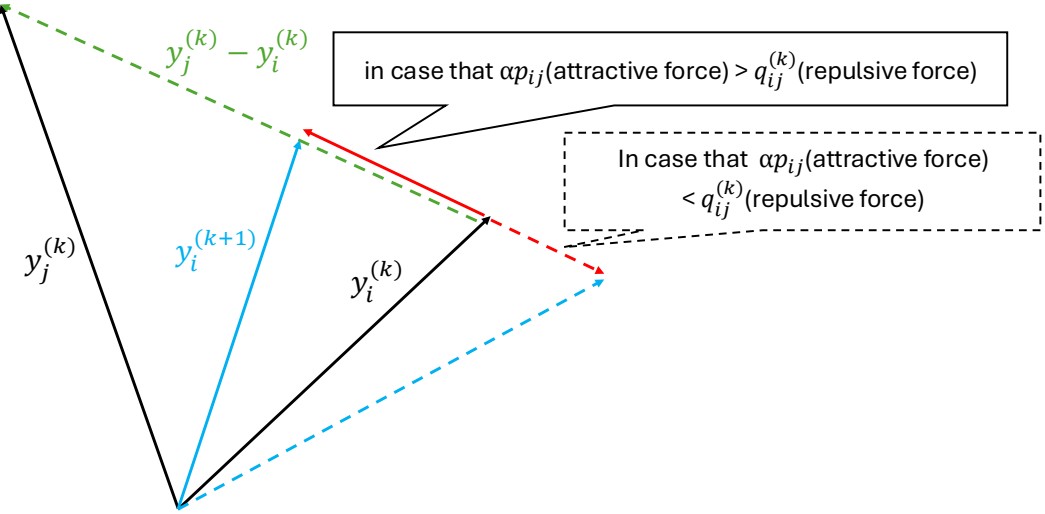

Figure 3: Diagram of the update equation for $y_i^{(k)}$

Therefore, the larger the value of $\alpha > 1$, the stronger the attractive force among nodes becomes. As a result, this generally facilitates the formation of clusters.

## B.2 The case of MM and NAG

As the update equations (10) for MM and (19) or (20) for NAG show, $y_i^{(k+1)}$ is calculated using both the gradient term and the momentum term. Acceleration is achieved by incorporating not only the current gradient but also past updates. Even when using MM or NAG, the amplification of the attractive force with $\alpha$ tends to occur.

In summary, when $\alpha > 1$, the attractive force generally becomes stronger, making it easier to form clusters across all optimization methods—GD, MM, and NAG. This suggests that a larger $\alpha$ can accelerate the clustering process.

## C Asymptotic behavior of the update equation

### C.1 Proof of Theorem 4.1

This theorem in Cai & Ma (2022) is based without momentum term in (3) and it can be extended even with momentum term, because the proof doesn't refer to the update equation (5), but just the definition of $S_{ij}^{(k)}(\alpha)$ in Eq. (4).

### C.2 Proof of Proposition 4.2

This proof proceeds in the same manner as the proof of Proposition 3 in Cai & Ma (2022). Note that $y_{li}^{(k+1)} \leq \|[\mathbf{I} - h\mathbf{S}_\alpha^{(k)}]_{i\cdot}\|_1 \|\mathbf{y}_l^{(k)}\|_\infty + m^{(k+1)}\|\mathbf{y}_l^{(k)} - \mathbf{y}_l^{(k-1)}\|_\infty$ for any $k \geq 1$, where

$$\|[\mathbf{I} - h\mathbf{S}_\alpha^{(k)}]_{i\cdot}\|_1 \|\mathbf{y}_l^{(k)}\|_\infty = \left|1 - h\sum_{j=1}^n S_{ij}^{(k)}(\alpha)\right| + h\sum_{j \neq i}|S_{ij}^{(k)}| \leq 1 + 2h\sum_{j=1}^n |S_{ij}^{(k)}(\alpha)|.$$

For the second term, we have

$$h\sum_{j=1}^n |S_{ij}^{(k)}(\alpha)| \leq hn\|\mathbf{S}_\alpha^{(k)}\|_\infty \leq nh(\alpha\|\mathbf{P}\|_\infty + \|\mathbf{Q}\|_\infty) \leq nh\alpha\|\mathbf{P}\|_\infty + \frac{h(1+\eta^{(k)})}{n-1},$$

where the last inequality follows from Eq. (40) in Cai & Ma (2022). Intermediate variable $Z$ is defined as $\sum_{i \neq j}(1 + \|y_i^{(k)} - y_j^{(k)}\|_2^2)^{-1}$ and we can deduce it with $\|\mathbf{Q}^{(k)}\|_\infty \leq 1/Z$, $Z \geq n(n-1)/(1+\eta^{(k)})$ and $\|\mathbf{Q}^{(k)}\|_\infty \leq (1+\eta^{(k)})/n(n-1)$. Then we have

$$y_{li}^{(k+1)} \leq \left(1 + 2nh\alpha\|\mathbf{P}\|_\infty + \frac{h(1+\eta^{(k)})}{n-1}\right)\|\mathbf{y}_l^{(k)}\|_\infty + m^{(k+1)}\|\mathbf{y}_l^{(k)}\|_\infty + m^{(k+1)}\|\mathbf{y}_l^{(k-1)}\|_\infty$$

$$\leq \left(1 + 2nh\alpha\|\mathbf{P}\|_\infty + \frac{h(1+\eta^{(k)})}{n-1} + m^{(k+1)}\right)\|\mathbf{y}_l^{(k)}\|_\infty + m^{(k+1)}\|\mathbf{y}_l^{(k-1)}\|_\infty$$

$$\leq \left(1 + 2nh\alpha\|\mathbf{P}\|_\infty + \frac{h(1+\eta^{(k)})}{n-1} + m^{(k+1)}\right)(\|\mathbf{y}_l^{(k)}\|_\infty + \|\mathbf{y}_l^{(k-1)}\|_\infty)$$

$$\leq \left(2 + 4nh\alpha\|\mathbf{P}\|_\infty + \frac{2h(1+\eta^{(k)})}{n-1} + 2m^{(k+1)}\right) \cdot \max_{l\in[2]}\{\|\mathbf{y}_l^{(k)}\|_\infty, \|\mathbf{y}_l^{(k-1)}\|_\infty\},$$

or

$$\|\mathbf{y}_l^{(k+1)}\|_\infty \leq \left(2 + 4nh\alpha\|\mathbf{P}\|_\infty + \frac{2h(1+\eta^{(k)})}{n-1} + 2m^{(k+1)}\right) \cdot \max_{l\in[2]}\{\|\mathbf{y}_l^{(k)}\|_\infty, \|\mathbf{y}_l^{(k-1)}\|_\infty\}.$$

Introducing $r_n = nh\alpha\|\mathbf{P}_\infty\|_\infty + \frac{h}{n}$, we have

$$\|\mathbf{y}_l^{(k+1)}\|_\infty \leq (2 + 2m^{(k+1)} + Cr_n) \cdot \max_{l\in[2]}\{\|\mathbf{y}_l^{(k)}\|_\infty, \|\mathbf{y}_l^{(k-1)}\|_\infty\} \tag{C.13}$$

and for any $k \geq 2$,

$$\|\mathbf{y}_l^{(k)}\|_\infty \leq (2 + 2m^{(k+1)} + Cr_n)^{k-1} \cdot \max_{l\in[2]}\{\|\mathbf{y}_l^{(1)}\|_\infty, \|\mathbf{y}_l^{(0)}\|_\infty\}. \tag{C.14}$$

**Lemma C.1.** *For $\forall k \geq 2$, $\eta^{(k)}$ and $\max_{l\in[2]}\{\|\mathbf{y}_l^{(k)}\|_\infty, \|\mathbf{y}_l^{(k-1)}\|_\infty\}$ are bounded by a constant.*

*Proof.* In case of $k = 2$, Eq. (C.14) shows that

$$\|\mathbf{y}_l^{(2)}\|_\infty \le (2 + 2m^{(k+1)} + Cr_n) \cdot \max_{l \in [2]}\{\|\mathbf{y}_l^{(1)}\|_\infty, \|\mathbf{y}_l^{(0)}\|_\infty\}. \tag{C.15}$$

The condition (T1) states that $\max_{l \in [2]}\{\|\mathbf{y}_l^{(2)}\|_\infty, \|\mathbf{y}_l^{(1)}\|_\infty\}$ is bounded. According to (C.14),

$$\eta^{(2)} \le 4(2 + 2m^{(2)} + Cr_n)^2 \cdot \max_{l \in [2]}\{\|\mathbf{y}_l^{(1)}\|_\infty^2, \|\mathbf{y}_l^{(0)}\|_\infty^2\} \tag{C.16}$$

is also bounded. Then, let's say that the statement holds for $k$, i.e. both $\|\mathbf{y}_l^{(k)}\|_\infty$ and $\|\mathbf{y}_l^{(k+1)}\|_\infty$ are bounded. With inequality (C.13), $\|\mathbf{y}_l^{(k+1)}\|_\infty$ is also bounded. Assuming $r_n = O(1)$ by condition (T1)

$$\eta^{(k+1)} \le 4 \max_{i \in [n], l \in [2]} |y_{li}^{(k+1)}|^2 \le 4 \max_{l \in [2]}\{\|\mathbf{y}_1^{(k+1)}\|_\infty^2, \mathbf{y}_2^{(k+1)}\|_\infty^2\} \tag{C.17}$$

$$\le 4(2 + 2m^{(k+1)} + Cr_n)^2 \cdot \max_{l \in [2]}\{\|\mathbf{y}_l^{(k)}\|_\infty^2, \|\mathbf{y}_l^{(k-1)}\|_\infty^2\} = O(1), \tag{C.18}$$

we have $\eta^{(k+1)}$ is bounded. By induction, the statement holds for $\forall k \ge 2$. $\qquad\square$

As long as $k = k(n)$ with $kr_n = O(1)$ by condition (T1), we have

$$\|\mathbf{y}_l^{(k)}\|_\infty / \max_{l \in [2]}\{\|\mathbf{y}_l^{(1)}\|_\infty, \|\mathbf{y}_l^{(0)}\|_\infty\} = O(1), \tag{C.19}$$

or,

$$\frac{\text{diam}(\{y_i^{(k)}\}_{i \in [n]})}{\max_{l \in [2]}\{\|\mathbf{y}_l^{(1)}\|_\infty, \|\mathbf{y}_l^{(0)}\|_\infty\}} \le \frac{\max_{i \in [n], l \in [2]} |y_{li}^{(k)}|}{\max_{l \in [2]}\{\|\mathbf{y}_l^{(1)}\|_\infty, \|\mathbf{y}_l^{(0)}\|_\infty\}} = O(1). \tag{C.20}$$

It proves the statement.

## D   Derivation of ODEs

We proceed to concretely examine the following two cases: MM and NAG.

### D.1   Proof of Lemma 5.1

The assumption $\tilde{\mathbf{y}}_l^{(k)} \approx \mathbf{Y}_l(kh)$ is justfied with the following satatement, which is shown similar to Proposition 8 in Cai & Ma (2022). We consider the momentum terms to be a constant $m (0 < m < 1)$, i.e. $m^{(k+1)} = m$ for $\forall k$. So, we can rewrite the formula (8) as follows:

$$\mathbf{y}_l^{(k+1)} \approx [\mathbf{I}_n - h\mathbf{L}(\alpha\mathbf{P} - \mathbf{H}_n)]\mathbf{y}_l^{(k)} + m(\mathbf{y}_l^{(k)} - \mathbf{y}_l^{(k-1)}), \quad l \in [2]. \tag{D.21}$$

For $l \in [2]$, let $\{\tilde{\mathbf{y}}_l^{(k)}\}_{k \ge 0}$ be the sequence defined the iteration with the update formula:

$$\tilde{\mathbf{y}}_l^{(k+1)} = [\mathbf{I}_n - h\mathbf{L}(\alpha\mathbf{P} - \mathbf{H}_n)]\tilde{\mathbf{y}}_l^{(k)} + m(\tilde{\mathbf{y}}_l^{(k)} - \tilde{\mathbf{y}}_l^{(k-1)}), \quad k \ge 1 \tag{D.22}$$

with initial values $\tilde{\mathbf{y}}_l^{(0)} = \mathbf{y}_l^{(0)}$, $\tilde{\mathbf{y}}_l^1 = \mathbf{y}_l^{(1)}$. Introduce the assumption $\mathbf{Y}_l(t) \approx \tilde{\mathbf{y}}_l^{(k)}$ for some smooth curve $Y_l(t)$ defined for $t \ge 0$. Set $t = kh$. Then, the update formula (D.22) reduces to

$$\mathbf{Y}_l(t + h) = \mathbf{Y}_l(t) - h\mathbf{L}(\alpha\mathbf{P} - \mathbf{H}_n)\mathbf{Y}_l(t) + m(\mathbf{Y}_l(t) - \mathbf{Y}_l(t - h)). \tag{D.23}$$

By dividing by $h > 0$, we have

$$\frac{\mathbf{Y}_l(t + h) - \mathbf{Y}_l(t)}{h} = -\mathbf{L}(\alpha\mathbf{P} - \mathbf{H}_n)\mathbf{Y}_l(t) + m \cdot \frac{\mathbf{Y}_l(t) - \mathbf{Y}_l(t - h)}{h}. \tag{D.24}$$

From the definition of the derivative $\frac{d}{dt}\mathbf{Y}_l(t) = \lim_{h \to 0} \frac{\mathbf{Y}_l(t+h) - \mathbf{Y}_l(t)}{h} = \lim_{h \to 0} \frac{\mathbf{Y}_l(t) - \mathbf{Y}_l(t-h)}{h}$, we have (15).

### D.2 Proof of Proposition 5.2

For any $k \in \mathbb{Z}_{\geq 0}$, we set $e_k = \tilde{\mathbf{y}}_l^{(k)} - \mathbf{Y}_l(kh)$, $\mathbf{L} = \mathbf{L}(\alpha \mathbf{P}_n - \mathbf{H}_n)$ for brevity.

**Lemma D.1.** *Let $\mathbf{Y}_l(t)$ the formula in Proposition 5.1, and a symmectic matrix $\mathbf{L}$ has an eigendecomposition,*

$$\mathbf{L} = Q\Lambda Q^{-1}, \tag{D.25}$$

*where $\Lambda = \mathrm{diag}(\lambda_1, \cdots, \lambda_n)$, $\lambda_i \in \mathbb{R}$, $\lambda_1 \leq \lambda_2 \leq \cdots \leq \lambda_n$ and $Q \in O(n)$. For $t > 0$*

$$\|\mathbf{Y}_l(t)\|_2 \leq \left\| \exp\left(-\frac{t\lambda_1}{1-m}\right) \right\| \cdot \|\mathbf{Y}_l(0)\|_2. \tag{D.26}$$

*Specifically, if $0 < m < 1$ and the condition (I1) holds, $\|\mathbf{Y}_l(t)\|_2$ is bounded.*

*Proof.* With the eigendecomposition $\mathbf{L} = Q\Lambda Q^{-1}$, we have

$$\|\mathbf{Y}_l(t)\|_2 \leq \left\| \exp\left(-\frac{t}{1-m}\mathbf{L}\right) \right\| \cdot \|\mathbf{Y}_l(0)\|_2, \tag{D.27}$$

$$\leq \left\| Q \exp\left(-\frac{t}{1-m}\Lambda\right) Q^{-1} \right\| \cdot \|\mathbf{Y}_l(0)\|_2, \tag{D.28}$$

$$\leq \left\| Q \right\| \left\| \exp\left(-\frac{t}{1-m}\Lambda\right) \right\| \left\| Q \right\|^{-1} \cdot \|\mathbf{Y}_l(0)\|_2, \tag{D.29}$$

$$\leq \left\| \exp\left(-\frac{t\lambda_1}{1-m}\right) \right\| \cdot \|\mathbf{Y}_l(0)\|_2. \tag{D.30}$$

$\square$

Using Taylor expansion for some $\xi \in (kh, (k+1)h)$, and Lemma 5.1, we have

$$\mathbf{Y}_l((k+1)h) = \mathbf{Y}_l(kh) + h\frac{d\mathbf{Y}_l(kh)}{dt} + \frac{h^2}{2}\frac{d^2\mathbf{Y}_l(\xi)}{dt^2} \tag{D.31}$$

$$= \mathbf{Y}_l(kh) - \frac{h}{1-m}\mathbf{L}\mathbf{Y}_l(kh) + \frac{h^2}{2}\left(\frac{1}{1-m}\right)^2 \mathbf{L}^2\mathbf{Y}_l(\xi). \tag{D.32}$$

We set these formulae into

$$e_{k+1} = \mathbf{Y}_l((k+1)h) - \tilde{\mathbf{y}}_l^{(k+1)}, \tag{D.33}$$

or,

$$e_{k+1} = \left(\mathbf{Y}_l(kh) - \frac{h}{1-m}\mathbf{L}\mathbf{Y}_l(kh) + \frac{1}{2}\left(\frac{h}{1-m}\right)^2 \mathbf{L}^2\mathbf{Y}_l(\xi)\right) \tag{D.34}$$

$$- \left((1+m)\mathbf{y}_l^{(k)} - h\mathbf{L}\tilde{\mathbf{y}}_l^{(k)} - m\tilde{\mathbf{y}}_l^{(k-1)}\right). \tag{D.35}$$

With $\mathbf{Y}_l(kh) = \tilde{\mathbf{y}}_l^{(k)} + e_k$, we have

$$e_{k+1} = e_k - \frac{hm}{1-m}\mathbf{L}\mathbf{Y}_l(kh) + m(\tilde{\mathbf{y}}_l^{(k-1)} - \tilde{\mathbf{y}}_l^{(k)}) + \frac{1}{2}\left(\frac{h}{1-m}\right)^2 \mathbf{L}^2\mathbf{Y}_l(\xi). \tag{D.36}$$

Due to triangular inequality, we have

$$\|e_{k+1}\|_2 \leq \|e_k\|_2 + \frac{hm}{1-m}\|\mathbf{L}\tilde{\mathbf{y}}_l^{(k)}\|_2 + m\|\tilde{\mathbf{y}}_l^{(k-1)} - \tilde{\mathbf{y}}_l^{(k)}\|_2 + \frac{1}{2}\left(\frac{h}{1-m}\right)^2 \|\mathbf{L}^2\mathbf{Y}_l(\xi)\|_2. \tag{D.37}$$

Summing up on the index, we obtain

$$\|e_{k+1}\|_2 \leq \|e_0\|_2 + \sum_{j=0}^{k}\left(\frac{hm}{1-m}\|\mathbf{L}\|\|\tilde{\mathbf{y}}_l^{(j)}\|_2 + m\|\tilde{\mathbf{y}}_l^{(j-1)} - \tilde{\mathbf{y}}_l^{(j)}\|_2 + \frac{1}{2}\left(\frac{h}{1-m}\right)^2 \|\mathbf{L}\|^2\|\mathbf{Y}_l(\xi_j)\|_2\right). \tag{D.38}$$

Thanks to our initial condition as in Lemma 5.1, $e_0 = \tilde{\mathbf{y}}_l^{(0)} - \mathbf{Y}_l(0) = (0, \cdots, 0)^\top$. Proposition 4.2 says that we have constant $C_1 > 0$ such that

$$\frac{\max_j \|\tilde{\mathbf{y}}_l^{(j)}\|_2}{\|\mathbf{Y}_l(0)\|_2} \le C_1. \tag{D.39}$$

Additionally, with respect to the term $\|\tilde{\mathbf{y}}_l^{(j-1)} - \tilde{\mathbf{y}}_l^{(j)}\|_2$, consider the update equation (10),

$$\tilde{\mathbf{y}}_l^{(j)} - \tilde{\mathbf{y}}_l^{(j-1)} = m(\tilde{\mathbf{y}}_l^{(j-1)} - \tilde{\mathbf{y}}_l^{(j-2)}) - h\mathbf{L}\tilde{\mathbf{y}}_l^{(j-1)}. \tag{D.40}$$

We have

$$\|\tilde{\mathbf{y}}_l^{(j)} - \tilde{\mathbf{y}}_l^{(j-1)}\|_2 \le m\|\tilde{\mathbf{y}}_l^{(j-1)} - \tilde{\mathbf{y}}_l^{(j-2)}\|_2 + hC_1\|\mathbf{L}\| \cdot \|\mathbf{Y}_l(0)\|_2. \tag{D.41}$$

By recursively applying these inequalities, we obtain

$$\|\tilde{\mathbf{y}}_l^{(j-1)} - \tilde{\mathbf{y}}_l^{(j)}\|_2 \le (m^j + m^{j-1} + \ldots + m + 1)hC_1\|\mathbf{L}\| \cdot \|\mathbf{Y}_l(0)\|_2 \tag{D.42}$$

$$= \frac{1 - m^j}{1 - m}hC_1\|\mathbf{L}\| \cdot \|\mathbf{Y}_l(0)\|_2 \le \frac{hC_1}{1 - m}\|\mathbf{L}\| \cdot \|\mathbf{Y}_l(0)\|_2. \tag{D.43}$$

With Lemma D.1 and (I1),

$$\frac{\|\mathbf{Y}_l(\xi_j)\|_2}{\|\mathbf{Y}_l(0)\|_2} \le \left\| \exp\left( -\frac{\xi_j \lambda_1}{1 - m} \right) \right\| \le \left\| \exp\left( -\frac{\xi_0 \lambda_1}{1 - m} \right) \right\|. \tag{D.44}$$

We denote the right-hand side as $C_2 > 0$. Under these conditions, we have

$$\frac{\|e_{k+1}\|_2}{\|\mathbf{Y}_l(0)\|_2} \le \frac{2hm(k+1)C_1}{1 - m}\|\mathbf{L}\| + \frac{k+1}{2}\left( \frac{h}{1 - m} \right)^2 \|\mathbf{L}\|^2 C_2. \tag{D.45}$$

This proves the statement.

### D.3 Proof of Lemma 5.3

Since

$$\mathbf{L}(\mathbf{P} - \alpha\mathbf{H}_n) = \mathbf{L}(\alpha\mathbf{P} - \mathbf{H}_n) - \frac{1}{n-1}\mathbf{I}_n + \frac{1}{n(n-1)}\mathbf{1}_n\mathbf{1}_n^\top, \tag{D.46}$$

the eigenvectors $\mathbf{L}(\alpha\mathbf{P} - \mathbf{H}_n)$ and $\mathbf{L}(\alpha\mathbf{P})$ share the eigenvectors. Also, if we decompose eigenvector as $\mathbf{L}(\mathbf{P} - \alpha\mathbf{H}_n) = \sum_{i=1}^n \sigma_i \mathbf{u}_i \mathbf{u}_i^\top$,

$$\mathbf{L}(\alpha\mathbf{P} - \mathbf{H}_n)u_i = \sigma_i u_i. \tag{D.47}$$

We have $\sigma_1 = \alpha\lambda_1, \sigma_i = \alpha\lambda_i - \frac{1}{n-1}$ for $2 \le i \le n$, which corresponds to (14).

### D.4 Proof of Lemma 5.5

Remember that both $\lambda_i(\mathbf{L}(\mathbf{P}^*)) \le \lambda_{i+1}(\mathbf{L}(\mathbf{P}^*))$ for $i \in [n-1]$ and $\lambda_{R+1}(\mathbf{L}(\mathbf{P}^*))$ is the first positive eigenvalue, i.e.

$$\lambda_R(\mathbf{L}(\mathbf{P}^*)) = 0, \lambda_{R+1}(\mathbf{L}(\mathbf{P}^*)) > 0. \tag{D.48}$$

With Weyl's inequality

$$|\lambda_i(\mathbf{L}(\mathbf{P})) - \lambda_i(\mathbf{L}(\mathbf{P}^*))| \le \|\mathbf{L}(\mathbf{E})\|, \tag{D.49}$$

where $\mathbf{E} = \mathbf{P} - \mathbf{P}^*$. When $i = R$ in inequality. (D.49), we have

$$|\alpha\lambda_R(\mathbf{L}(\mathbf{P}))| \le \alpha\|\mathbf{L}(\mathbf{E})\|. \tag{D.50}$$

The condition (T2.M) says that $\frac{t}{1-m}\alpha\|\mathbf{L}(\mathbf{E})\| = o(1)$, and it's equivalent that

$$\forall\epsilon > 0, \exists N \in \mathbb{N} \text{ s.t. } n \ge N \Rightarrow \frac{t}{1 - m}\alpha\|\mathbf{L}(\mathbf{E})\| < \epsilon. \tag{D.51}$$

If we set $\epsilon = \frac{t}{(n-1)(1-m)}$, we have

$$\alpha\lambda_R(\mathbf{L}(\mathbf{P})) - \frac{1}{n-1} \leq \alpha\|\mathbf{L}(\mathbf{E})\| - \frac{1}{n-1} \tag{D.52}$$

$$< \frac{(1-m)\epsilon}{t} - \frac{1}{n-1} \tag{D.53}$$

$$< \frac{1}{t} \times (1-m) \times \frac{t}{(n-1)(1-m)} - \frac{1}{n-1} = 0. \tag{D.54}$$

By contrast, when $i = R+1$ in Ineq. (D.49)

$$|\lambda_{R+1}(\mathbf{L}(\mathbf{P})) - \lambda_{R+1}(\mathbf{L}(\mathbf{P}^*))| \leq \|\mathbf{L}(\mathbf{E})\|. \tag{D.55}$$

So, we have

$$\lambda_{R+1}(\mathbf{L}(\mathbf{P})) \geq \lambda_{R+1}(\mathbf{L}(\mathbf{P}^*)) - \|\mathbf{L}(\mathbf{E})\|. \tag{D.56}$$

If we set $\epsilon = (1 - \frac{t}{n-1}) \times \frac{1}{1-m}$ and the condition (T2.M) such as $\lambda_{R+1}(\mathbf{L}(\mathbf{P}^*)) \gg (t\alpha)^{-1}$

$$\alpha\lambda_{R+1}(\mathbf{L}(\mathbf{P})) - \frac{1}{n-1} \geq \alpha\lambda_{R+1}(\mathbf{L}(\mathbf{P}^*)) - \alpha\|\mathbf{L}(\mathbf{E})\| - \frac{1}{n-1} \tag{D.57}$$

$$> \alpha\lambda_{R+1}(\mathbf{L}(\mathbf{P}^*)) - \frac{(1-m)\epsilon}{t} - \frac{1}{n-1} \tag{D.58}$$

$$> \alpha \times \frac{1}{t\alpha} - \frac{1-m}{t} \times \left(1 - \frac{t}{n-1}\right) \times \frac{1}{1-m} - \frac{1}{n-1} = 0. \tag{D.59}$$

Note that the condition (T1.M) $t = o(n)$ assures that $\epsilon = (1 - \frac{t}{n-1}) \times \frac{1}{1-m} > 0$ for $n \gg 1$ and $0 < m < 1$.

## E  Well-conditioned matrix

As discussed in section 5.1, we're interested in the behavior of $\mathbf{Y}_l(t)$ for $t \gg 1$. The sign of the terms $\alpha\lambda_i - \frac{1}{n-1}$ decides it; if negative, the corresponding components are amplified; if positive, they are suppressed as noted in (17).

This scenario, where $\alpha\lambda_i - \frac{1}{n-1} \leq 0$ for $2 \leq i \leq R$, indicates that the $i$-th eigenvalue $\lambda_i$ is close to 0 under $\alpha > 0$. Although the matrix $\mathbf{L}(\mathbf{P})$, by definition, has one connected component, suppose there exists another matrix $\mathbf{P}^*$ sufficiently close to $\mathbf{P}$ such that its Laplacian $\mathbf{L}(\mathbf{P}^*)$ has $R$ connected components, implying the dimension of its null space is $R$. As discussed in (Cai & Ma, 2022), it is natural to pick up Laplacian null space. We call the adjacency matrix $\mathbf{P}^*$ as "well-conditioned", if its associated weighted graph has $R \geq 2$ connected components.

**Proposition E.1.** *(Proposition 6 in (Cai & Ma, 2022)) Let $\mathbf{A} \in \mathbb{R}^{n \times n}$ be symmetric and well-conditioned matrix. Then, the smallest eigenvalue of the Laplacian $\mathbf{L}(\mathbf{A})$ is 0 and has multiplicity $R$, and its associated eigenspace is spanned by $\{\boldsymbol{\theta}_1, \ldots, \boldsymbol{\theta}_R\}$, where for each $r \in [R]$,*

$$[\boldsymbol{\theta}_i]_j = \begin{cases} 1/\sqrt{n_i}, & \text{if node } j \text{ belongs component } i, \\ 0, & \text{otherwise}, \end{cases} \tag{E.60}$$

*and $n_r$ is the number of nodes related to $r$-th connected component. Also, the null space of $\mathbf{L}(\mathbf{A})$ is spanned with these basis*

$$\frac{1}{\sqrt{n_1}}\begin{bmatrix} \mathbf{1}_{n_1} \\ \mathbf{0} \\ \vdots \\ \mathbf{0} \end{bmatrix}, \frac{1}{\sqrt{n_2}}\begin{bmatrix} \mathbf{0} \\ \mathbf{1}_{n_2} \\ \vdots \\ \mathbf{0} \end{bmatrix}, \ldots, \frac{1}{\sqrt{n_R}}\begin{bmatrix} \mathbf{0} \\ \vdots \\ \mathbf{0} \\ \mathbf{1}_{n_R} \end{bmatrix}, \tag{E.61}$$

*where $\mathbf{1}_{n_r}$ is a length $n_r$ column vector with ones, and $\mathbf{0}$ is a zero vector with appropriate length.*

When the data $\{X_i\}_{i \in [n]}$ are well clustered and bandwidths $\tau_i$ are appropriately selected, in (Balakrishnan et al., 2011), it is shown that we have a good approximated matrix of $\mathbf{P}$. Such well-conditioned matrix $\mathbf{P}^*$ is obtained concretely in section 8. Such matrix $\mathbf{P}^*$ is theoretically used in implicit regularization theorem 6.5.

# F Derivation of ODE for NAG

## F.1 Proof of Lemma 6.1

With two equations (19) and (20), we derive another ODE. Applying a rescaling, we have

$$\frac{\tilde{\mathbf{y}}_l^{(k+1)} - \tilde{\mathbf{y}}_l^{(k)}}{\sqrt{h}} = \frac{k-1}{k+2} \frac{\tilde{\mathbf{y}}_l^{(k)} - \tilde{\mathbf{y}}_l^{(k-1)}}{\sqrt{h}} - \sqrt{h}\mathbf{L}(\alpha\mathbf{P} - \mathbf{H}_n)\mathbf{w}_l^{(k)}. \tag{F.62}$$

Introduce the [assumption] $\tilde{\mathbf{y}}_l^{(k)} \approx \mathbf{Y}(k\sqrt{h})$ for some smooth curve $\mathbf{Y}_l(t)$ for $t \geq 0$. Set $k = t/\sqrt{h}$. Then, as the step size $h$ goes to zero, $\mathbf{Y}_l(t) \approx \tilde{\mathbf{y}}_l^{(t/\sqrt{h})} = \tilde{\mathbf{y}}_l^{(k)}$ and $\mathbf{Y}_l(t + \sqrt{h}) \approx \tilde{\mathbf{y}}_l^{((t+\sqrt{h})/\sqrt{h})} = \tilde{\mathbf{y}}_l^{(k+1)}$ Taylor expansion gives

$$(\mathbf{y}_l^{(k+1)} - \mathbf{y}_l^{(k)})/\sqrt{h} = \dot{\mathbf{Y}}_l(t) + \frac{1}{2}\ddot{\mathbf{Y}}_l(t)\sqrt{h} + o(\sqrt{h}), (\mathbf{y}_l^{(k)} - \mathbf{y}_l^{(k-1)})/\sqrt{h} = \dot{\mathbf{Y}}_l(t) - \frac{1}{2}\ddot{\mathbf{Y}}_l(t)\sqrt{h} + o(\sqrt{h})$$

and $\sqrt{h}\mathbf{L}(\alpha\mathbf{P} - \mathbf{H}_n)w_l^{(k)} = \sqrt{h}\mathbf{L}(\alpha\mathbf{P} - \mathbf{H}_n)\mathbf{Y}_l + o(\sqrt{h})$. So, (F.62) is written as

$$\dot{\mathbf{Y}}_l(t) + \frac{1}{2}\ddot{\mathbf{Y}}_l(t)\sqrt{h} + o(\sqrt{h}) = \left(1 - \frac{3\sqrt{h}}{t}\right)\left(\dot{\mathbf{Y}}_l(t) - \frac{1}{2}\ddot{\mathbf{Y}}_l(t)\sqrt{h} + o(\sqrt{h})\right) - \sqrt{h}\mathbf{L}(\alpha\mathbf{P} - \mathbf{H}_n)\mathbf{Y}_l(t) + o(\sqrt{h}).$$

By comparing the coefficients of $\sqrt{h}$, we achieve

$$\ddot{\mathbf{Y}}_l(t) + \frac{3}{t}\dot{\mathbf{Y}}_l(t) + \mathbf{L}(\alpha\mathbf{P} - \mathbf{H}_n)\mathbf{Y}_l(t) = 0, \quad l \in [2]. \tag{F.63}$$

Note that the first initial condition is $\mathbf{Y}_l(0) = \mathbf{y}_l^{(0)}$. Also, taking $k = 1$ in (F.62), we have

$$(\mathbf{y}_l^{(1)} - \mathbf{y}_l^{(0)})/\sqrt{h} = -\sqrt{h}\mathbf{L}(\alpha\mathbf{P} - \mathbf{H}_n)\mathbf{w}_l(t) = o(1).$$

Then, the second initial condition is just $\dot{\mathbf{Y}}_l(0) = (0, \ldots, 0)^\top \in \mathbb{R}^n$.

## F.2 Proof of Proposition 6.2

If we define a function $f : \mathbb{R}^n \to \mathbb{R}^n$ as $f(y) = \frac{1}{2}y^\top \mathbf{L}(\alpha\mathbf{P} - \mathbf{H}_n)y$ for $y \in \mathbb{R}^n$, the function $f$ is differentiable and $\nabla f(y) = \mathbf{L}(\alpha\mathbf{P} - \mathbf{H}_n)y$. So, we have

$$\|\nabla f(x) - \nabla f(y)\|_2 \leq \sigma_n \|x - y\|_2, \tag{F.64}$$

where $\sigma_n$ is the maximum eigenvalue of $\mathbf{L}(\alpha\mathbf{P} - \mathbf{H}_n)$, especially $f$ is $\sigma_n$-Lipschitz function. So, we can apply Proposition 2 in Su et al. (2016), which shows our statement.

## F.3 Proof of Lemma 6.3

Similar to Lemma 5.5, we can deduce the results since we have $t^2\|\mathbf{L}(\mathbf{P} - \mathbf{P}^*)\| = o(1)$, that is

$$\forall \epsilon > 0, \exists N \in \mathbb{N} \text{ s.t. } n \geq N \Rightarrow t^2\alpha\|\mathbf{L}(\mathbf{E})\| < \epsilon, \tag{F.65}$$

$\lambda_{R+1}(\mathbf{L}(\mathbf{P}^*)) \gg (t^2\alpha)^{-1}$ which are both in (T2.N) and $t = o(n^{1/2})$ as in (T1.N).

## F.4 Proof of Proposition 6.4

As in the Lemma 5.3, we have $\mathbf{L}(\alpha\mathbf{P} - \mathbf{H}_n) = \mathbf{V}\boldsymbol{\Sigma}\mathbf{V}^\top$

$$\frac{d^2\mathbf{Y}_l(t)}{dt} + \frac{3}{t}\frac{d\mathbf{Y}_l(t)}{dt} + \mathbf{V}\boldsymbol{\Sigma}\mathbf{U}^\top\mathbf{Y}_l(t) = 0. \tag{F.66}$$

By multiplying $\mathbf{V}^\top$ from left-hand side, we have

$$\frac{d^2}{dt^2}(\mathbf{V}^\top \mathbf{Y}_l(t)) + \frac{3}{t}\frac{d}{dt}(\mathbf{V}^\top \mathbf{Y}_l(t)) + \Sigma(\mathbf{V}^\top \mathbf{Y}_l(t)) = 0, \tag{F.67}$$

where we used $\mathbf{V}^\top \mathbf{V} = I_n$. By replacing $\mathbf{V}^\top \mathbf{Y}_l(t)$ with $\mathbf{Y}_l(t)$,

$$\frac{d^2\mathbf{Y}_l(t)}{dt^2} + \frac{3}{t}\frac{d\mathbf{Y}_l(t)}{dt} + \Sigma\mathbf{Y}_l(t) = 0, \tag{F.68}$$

$$\frac{d^2\mathbf{Y}_l(t)}{dt^2} + \frac{3}{t}\frac{d\mathbf{Y}_l(t)}{dt} + \begin{pmatrix} \sigma_1 & & \\ & \ddots & \\ & & \sigma_n \end{pmatrix}\mathbf{Y}_l(t) = 0. \tag{F.69}$$

This means that for any $i \in [n]$,

$$\frac{d^2\mathbf{Y}_{l,i}(t)}{dt^2} + \frac{3}{t}\frac{d\mathbf{Y}_{l,i}(t)}{dt} + \sigma_i\mathbf{Y}_{l,i}(t) = 0 \tag{F.70}$$

with $\mathbf{Y}_{l,i}(0) = \mathbf{y}_{l,i}^{(0)}, \dot{\mathbf{Y}}_{l,i}(0) = 0$.

**Lemma F.1.** *For modified Bessel function of the first kind $I_1(\cdot)$, we have*

$$\lim_{s\to+0} \frac{2I_1(s)}{s} = 1. \tag{F.71}$$

*Proof.* With Taylor expansion around 0, we have

$$I_1(s) = \sum_{k=0}^{\infty} \frac{1}{k!\Gamma(k+2)}\left(\frac{t}{2}\right)^{2k+1}, \tag{F.72}$$

where $\Gamma(k+2)$ is Gamma function at $k+2$. For $s > 0$,

$$\frac{I_1(s)}{s} = \frac{1}{2}\sum_{k=0}^{\infty} \frac{1}{k!(k+1)!}\left(\frac{s}{2}\right)^{2k}. \tag{F.73}$$

Then,

$$\lim_{s\to+0} \frac{I_1(s)}{s} = \frac{1}{2}\cdot\frac{1}{0!1!} = \frac{1}{2}. \tag{F.74}$$

$\square$

**Proposition F.2.** *ODE (F.70) has the following explicit expressions:*

$$\mathbf{Y}_{l,i}(t) = \begin{cases} \dfrac{2\mathbf{y}_{l,i}^{(0)}}{t\sqrt{\sigma_i}} J_1(t\sqrt{\sigma_i}), & \text{if } \sigma_i > 0, \\[2ex] \dfrac{2\mathbf{y}_{l,i}^{(0)}}{t\sqrt{-\sigma_i}} I_1(t\sqrt{-\sigma_i}), & \text{if } \sigma_i \leq 0, \end{cases} \tag{F.75}$$

*where $J_1(\cdot)$ is Bessel function of the first kind, $I_1(\cdot)$ is the modified Bessel function of the first kind.*

*Proof.* Firstly, consider the case of $\sigma > 0$. For simplicity, we omit the indices $l$ and $Z_i(u) = u\mathbf{Y}_i(u/\sqrt{\sigma_i})$ which satisfies

$$u^2\ddot{Z}_i + u\dot{Z}_i + (u^2 - 1)Z_i = 0.$$

When we set $v = u/\sqrt{\sigma_i}$, we have

$$\frac{dZ_i}{du} = \frac{d}{du}\Big(u\mathbf{Y}_i\Big(\frac{u}{\sqrt{\sigma_i}}\Big)\Big) = \frac{d}{du}\Big(u\mathbf{Y}_i(v)\Big) = \mathbf{Y}_i(v) + v\dot{\mathbf{Y}}_i(v),$$

$$\frac{d^2Z_i}{du^2} = \frac{d}{du}\Big(\mathbf{Y}_i\Big(\frac{u}{\sqrt{\sigma_i}}\Big) + \frac{u}{\sqrt{\sigma_i}}\dot{\mathbf{Y}}_i\Big(\frac{u}{\sqrt{\sigma_i}}\Big)\Big) = \frac{2}{\sqrt{\sigma_i}}\dot{\mathbf{Y}}_i\Big(\frac{u}{\sqrt{\sigma_i}}\Big) + \frac{u}{\sigma_i}\ddot{\mathbf{Y}}_i\Big(\frac{u}{\sqrt{\sigma_i}}\Big).$$

So, we proceed the calculation

$$u^2\ddot{Z}_i + u\dot{Z}_i + (u^2 - 1)Z_i$$
$$= u^2\Big\{\frac{2}{\sqrt{\sigma_i}}\dot{\mathbf{Y}}_i\Big(\frac{u}{\sqrt{\sigma_i}}\Big) + \frac{u}{\sigma_i}\ddot{\mathbf{Y}}_i\Big(\frac{u}{\sqrt{\sigma_i}}\Big)\Big\} + u\Big\{\mathbf{Y}_i\Big(\frac{u}{\sqrt{\sigma_i}}\Big) + \frac{u}{\sqrt{\sigma_i}}\dot{\mathbf{Y}}_i\Big(\frac{u}{\sqrt{\sigma_i}}\Big)\Big\} + (u^2 - 1)u\mathbf{Y}_i\Big(\frac{u}{\sqrt{\sigma_i}}\Big)$$
$$= \frac{2u^2}{\sqrt{\sigma_i}}\dot{\mathbf{Y}}_i\Big(\frac{u}{\sqrt{\sigma_i}}\Big) + \frac{u^3}{\sigma_i}\ddot{\mathbf{Y}}_i\Big(\frac{u}{\sqrt{\sigma_i}}\Big) + u\mathbf{Y}_i\Big(\frac{u}{\sqrt{\sigma_i}}\Big) + \frac{u^2}{\sqrt{\sigma_i}}\dot{\mathbf{Y}}_i\Big(\frac{u}{\sqrt{\sigma_i}}\Big) + u^3\mathbf{Y}_i\Big(\frac{u}{\sqrt{\sigma_i}}\Big) - u\mathbf{Y}_i\Big(\frac{u}{\sqrt{\sigma_i}}\Big)$$
$$= \frac{u^3}{\sigma_i}\Big\{\ddot{\mathbf{Y}}_i\Big(\frac{u}{\sqrt{\sigma_i}}\Big) + \frac{3\sqrt{\sigma_i}}{u}\dot{\mathbf{Y}}_i\Big(\frac{u}{\sqrt{\sigma_i}}\Big) + \sigma_i\mathbf{Y}_i\Big(\frac{u}{\sqrt{\sigma_i}}\Big)\Big\} = 0.$$

The solution function $Z_i(u)$ can be expressed with $J_1(u)$

$$J_1(u) = \sum_{m=0}^{\infty} \frac{(-1)^m}{(2m)!!(2m+2)!!} u^{2m+1},$$

where we get around

$$J_1(u) = (1 + o(1))\frac{u}{2} \tag{F.76}$$

near zero for the variable $u$. Remember that $\mathbf{Y}_{l,i}^{(0)} = \mathbf{y}_{l,i}^{(0)}$, we have

$$\mathbf{Y}_{l,i}(t) = \frac{2\mathbf{y}_{l,i}^{(0)}}{t\sqrt{\sigma_i}}J_1(t\sqrt{\sigma_i}), \text{ for } \sigma_i > 0. \tag{F.77}$$

Secondly, we address the case of $\sigma_i < 0$. $Z_i(u) = u\mathbf{Y}_{l,i}(u/\sqrt{-\sigma_i})$, and we obtain the following modified Bessel differential equation:

$$u^2\ddot{Z}_i + u\dot{Z}_i - (u^2 + 1)Z_i = 0. \tag{F.78}$$

This also comes from the following calculation:

$$\frac{d\mathbf{Y}_i}{du} = \frac{1}{\sqrt{-\sigma_i}}\dot{\mathbf{Y}}\Big(\frac{u}{\sqrt{-\sigma_i}}\Big), \tag{F.79}$$

$$\frac{dZ_i}{du} = \frac{d}{du}\Big(u\mathbf{Y}_i\Big(\frac{u}{\sqrt{-\sigma_i}}\Big)\Big) = \mathbf{Y}_i\Big(\frac{u}{\sqrt{-\sigma_i}}\Big) + \frac{u}{\sqrt{-\sigma_i}}\dot{\mathbf{Y}}_i\Big(\frac{u}{\sqrt{-\sigma_i}}\Big), \tag{F.80}$$

$$\frac{d^2Z_i}{du^2} = \frac{d}{du}\Big(\mathbf{Y}_i\Big(\frac{u}{\sqrt{-\sigma_i}}\Big) + \frac{u}{\sqrt{-\sigma_i}}\dot{\mathbf{Y}}_i\Big(\frac{u}{\sqrt{-\sigma_i}}\Big)\Big) = \frac{2}{\sqrt{-\sigma_i}}\dot{\mathbf{Y}}_i\Big(\frac{u}{\sqrt{-\sigma_i}}\Big) - \frac{u}{\sigma_i}\ddot{\mathbf{Y}}_i\Big(\frac{u}{\sqrt{-\sigma_i}}\Big). \tag{F.81}$$

Then, we continue

$$u^2\ddot{Z}_i + u\dot{Z}_i - (u^2 + 1)Z_i \tag{F.82}$$

$$= u^2\Big\{\frac{2}{\sqrt{-\sigma_i}}\dot{\mathbf{Y}}_i\Big(\frac{u}{\sqrt{-\sigma_i}}\Big) - \frac{u}{\sigma_i}\ddot{\mathbf{Y}}_i\Big(\frac{u}{\sqrt{-\sigma_i}}\Big)\Big\} + u\Big\{\mathbf{Y}_i\Big(\frac{u}{\sqrt{-\sigma_i}}\Big) + \frac{u}{\sqrt{-\sigma_i}}\dot{\mathbf{Y}}_i\Big(\frac{u}{\sqrt{-\sigma_i}}\Big)\Big\} \tag{F.83}$$

$$- (u^2 + 1)u\mathbf{Y}_i\Big(\frac{u}{\sqrt{-\sigma_i}}\Big) \tag{F.84}$$

$$= -\frac{u^3}{\sigma_i}\ddot{\mathbf{Y}}_i\Big(\frac{u}{\sqrt{-\sigma_i}}\Big) + \frac{3u^2}{\sqrt{-\sigma_i}}\dot{\mathbf{Y}}_i\Big(\frac{u}{\sqrt{-\sigma_i}}\Big) - u^3\mathbf{Y}_i\Big(\frac{u}{\sqrt{-\sigma_i}}\Big) \tag{F.85}$$

$$= -\frac{u^3}{\sigma_i}\Big\{\ddot{\mathbf{Y}}_i\Big(\frac{u}{\sqrt{-\sigma_i}}\Big) + \frac{3\sqrt{-\sigma_i}}{u}\dot{\mathbf{Y}}_i\Big(\frac{u}{\sqrt{-\sigma_i}}\Big) + \sigma_i\mathbf{Y}_i\Big(\frac{u}{\sqrt{-\sigma_i}}\Big)\Big\} = 0. \quad (\because (\sqrt{-\sigma_i})^2 = -\sigma_i). \tag{F.86}$$

Similar to the case of $\sigma_i > 0$, we have the explicit expression of $Z_i(u)$ with $I_1(u)$

$$I_1(u) = \sum_{m=0}^{\infty} \frac{1}{m!\,(m+1)!} \left(\frac{u}{2}\right)^{2m+1},$$

or,

$$\mathbf{Y}_{l,i}(t) = \frac{2\mathbf{y}_{l,i}^{(0)}}{t\sqrt{-\sigma_i}} I_1(t\sqrt{-\sigma_i}), \text{ for } \sigma_i < 0. \tag{F.87}$$

This formula can be extended under $\sigma = 0$ with the formula (F.71) i.e. $\lim_{u \to 0} I_1(u)/u = \mathbf{y}_{l,i}^{(0)}$.

$\square$

As in Lemma 6.3, $\sigma_i$ are $R$-nonpositive eigenvalues, and there are $(n - R)$-positive eigenvalues. So, $\mathbf{Y}_l(t)$ was calculated as $\mathbf{V}^\top \mathbf{Y}_l(t)$, and multiplying $\mathbf{V}(= (\mathbf{U}, \mathbf{U}_\perp))$, we finally obtain

$$\mathbf{Y}_l(t) = \mathbf{V}\Big(y_{l,1}^{(0)}, \frac{2y_{l,2}^{(0)}}{t\sqrt{-\sigma_2}} I_1(t\sqrt{-\sigma_2}), \ldots, \frac{2y_{l,R}^{(0)}}{t\sqrt{-\sigma_R}} I_1(t\sqrt{-\sigma_R}),$$

$$\frac{2y_{l,R+1}^{(0)}}{t\sqrt{\sigma_{R+1}}} J_1(t\sqrt{\sigma_{R+1}}), \ldots, \frac{2y_{l,n}^{(0)}}{t\sqrt{\sigma_n}} J_1(t\sqrt{\sigma_n})\Big) \tag{F.88}$$

$$= (u_1, \ldots, u_R)\mathrm{diag}\Big(1, \frac{2}{t\sqrt{-\sigma_2}} I_1(t\sqrt{-\sigma_2}), \ldots, \frac{2}{t\sqrt{-\sigma_R}} I_1(t\sqrt{-\sigma_R})\Big)(y_{l,1}^{(0)}, \ldots, y_{l,R}^{(0)}) \tag{F.89}$$

$$+ (u_{R+1}, \ldots, u_n)\mathrm{diag}\Big(\frac{2}{t\sqrt{\sigma_{R+1}}} J_1(t\sqrt{\sigma_{R+1}}), \ldots, \frac{2}{t\sqrt{\sigma_n}} J_1(t\sqrt{\sigma_n})\Big)(y_{l,R+1}^{(0)}, \ldots, y_{l,n}^{(0)}) \tag{F.90}$$

$$= \mathbf{U}\boldsymbol{\Gamma}_1(t)\mathbf{y}_{l,1:R}^{(0)} + \mathbf{U}_\perp \boldsymbol{\Gamma}_2(t)\mathbf{y}_{l,R+1:n}^{(0)}, \tag{F.91}$$

where $\mathbf{U} = (u_1, \ldots, u_R) \in O(n, R)$ and $\mathbf{U}_\perp = (u_{R+1}, \ldots, u_n) \in O(n, n - R)$ is orthogonal complement of $\mathbf{U}$. Also

$$\boldsymbol{\Gamma}_1(t) = \mathrm{diag}\Big(1, \frac{2}{t\sqrt{-\sigma_2}} I_1(t\sqrt{-\sigma_2}), \ldots, \frac{2}{t\sqrt{-\sigma_R}} I_1(t\sqrt{-\sigma_R})\Big), \tag{F.92}$$

$$\boldsymbol{\Gamma}_2(t) = \mathrm{diag}\Big(\frac{2}{t\sqrt{\sigma_{R+1}}} J_1(t\sqrt{\sigma_{R+1}}), \ldots, \frac{2}{t\sqrt{\sigma_n}} J_1(t\sqrt{\sigma_n})\Big). \tag{F.93}$$

It proves the proposition.

# G    Implicit regularization

## G.1    Proof of Proposition 6.5 concerning MM

The proof for the MM can be directly adapted from proof of Theorem 10 in Cai & Ma (2022), requiring only the substitution of $e^{-t(\alpha\lambda_i - \frac{1}{n-1})}$ for $e^{-\frac{t}{1-m}(\alpha\lambda_i - \frac{1}{n-1})}$.

## G.2    Proof of Proposition 6.5 concerning NAG

Before proceeding with the proof, let us first establish the following lemma:

**Lemma G.1.**    *1. For orthonormal basis $\mathbf{U} \in O(n, R)$, we can define $\mathbf{U}_\perp \in O(n, n - R)$ and obtain*

$$\|\mathbf{U}\| = \|\mathbf{U}_\perp\| = 1. \tag{G.94}$$

*2. We have the following asymptotic approximation with $J_1(s)$ for $s \gg 1$,*

$$\frac{1}{s} J_1(s) \sim \sqrt{\frac{2}{\pi s^3}} \cos\Big(s - \frac{3\pi}{4}\Big). \tag{G.95}$$

3. *Similarly, for modified Bessel function $I_1(s)$, we have the asymptotic approximation for $s \gg 1$,*

$$I_1(s) \sim \frac{1}{\sqrt{2\pi s}} e^s. \tag{G.96}$$

*Proof.*  1. For (G.94), we obtain the results with basic linear algebra.

2. The asymptotic expansion of $J_1(s)$ is found in 7.21 of Watson (1922) or 10.7.8 in Olver & Maximon (2010) such as

$$J_1(s) \sim \sqrt{\frac{2}{\pi s}} \cos\left(s - \frac{3\pi}{4}\right). \tag{G.97}$$

Dividing by $s > 0$, we have the formula.

3. The statement follows from 7.23 of Watson (1922).

$\square$

The basic strategy on the proof for NAG still aligns to Theorem 10 in Cai & Ma (2022). Let $\mathbf{U}_0 \in O(n, R)$ be the matrix whose columns span the null space of $\mathbf{L}(\mathbf{P}^*)$, and the first column of $\mathbf{U}_0$ is $n^{-1/2}\mathbf{1}$. Also, let $\mathbf{U} \in O(n, R)$ be the collection of eigenvectors of $\mathbf{L}(\mathbf{P})$ corresponding the smallest $R$ eigenvalues. With the Davis-Kahan theorem discussed in von Luxburg (2007); Yu et al. (2015), we have

$$\|\mathbf{U}_{0\perp}^\top \mathbf{U}\| = \|\mathbf{U}_0^\top \mathbf{U}_\perp\| \le \frac{\|\mathbf{L}(\mathbf{E})\|}{\lambda_{R+1}(\mathbf{L}(\mathbf{P}^*))}, \tag{G.98}$$

where $\mathbf{E} = \mathbf{P} - \mathbf{P}^*$ and $\mathbf{U}_{0\perp}$ is orthogonal complement of $\mathbf{U}_0$.

$$\frac{\|\mathbf{U}_0 \mathbf{U}_0^\top \mathbf{Y}_l(t) - \mathbf{Y}_l(t)\|_2}{\|\mathbf{Y}_l(0)\|_2} \tag{G.99}$$

$$= \frac{\|\mathbf{U}_0 \mathbf{U}_0^\top \mathbf{U}\mathbf{\Gamma}_1(t)\mathbf{y}_{l,1:R}^{(0)} - \mathbf{U}\mathbf{\Gamma}_1(t)\mathbf{y}_{l,1:R}^{(0)}\|_2}{\|\mathbf{Y}_l(0)\|_2} + \frac{\|\mathbf{U}_0 \mathbf{U}_0^\top \mathbf{U}_\perp \mathbf{\Gamma}_2(t)\mathbf{y}_{l,1:R}^{(0)} - \mathbf{U}_\perp \mathbf{\Gamma}_2(t)\mathbf{y}_{l,R+1:n}^{(0)}\|_2}{\|\mathbf{Y}_l(0)\|_2} \tag{G.100}$$

$$= \frac{\|(\mathbf{U}_0 \mathbf{U}_0^\top - \mathbf{I})\mathbf{U}\mathbf{\Gamma}_1(t)\mathbf{y}_{l,1:R}^{(0)}\|_2}{\|\mathbf{Y}_l(0)\|_2} + \frac{\|(\mathbf{U}_0 \mathbf{U}_0^\top - \mathbf{I})\mathbf{U}_\perp \mathbf{\Gamma}_2(t)\mathbf{y}_{l,1:R}^{(0)}\|_2}{\|\mathbf{Y}_l(0)\|_2} \tag{G.101}$$

$$= \frac{\|(\mathbf{U}_{0\perp} \mathbf{U}_{0\perp}^\top)\mathbf{U}\mathbf{\Gamma}_1(t)\mathbf{y}_{l,1:R}^{(0)}\|_2}{\|\mathbf{Y}_l(0)\|_2} + \frac{\|(\mathbf{U}_{0\perp} \mathbf{U}_{0\perp}^\top)\mathbf{U}_\perp \mathbf{\Gamma}_2(t)\mathbf{y}_{l,1:R}^{(0)}\|_2}{\|\mathbf{Y}_l(0)\|_2} \tag{G.102}$$

$$\le \|\mathbf{U}_{0\perp}(\mathbf{U}_{0\perp}^\top \mathbf{U})\mathbf{\Gamma}_1(t)\| + \|(\mathbf{U}_{0\perp} \mathbf{U}_{0\perp}^\top)\mathbf{U}\mathbf{\Gamma}_2(t)\| \tag{G.103}$$

$$\le \|\mathbf{U}_{0\perp}^\top \mathbf{U}\| \cdot \|\mathbf{\Gamma}_1(t)\| + \|\mathbf{\Gamma}_2(t)\| \tag{G.104}$$

$$\le \frac{\|\mathbf{L}(\mathbf{E})\|}{\lambda_{R+1}(\mathbf{L}(\mathbf{P}^*))} \cdot \frac{2}{t\sqrt{-\sigma_R}} I_1(t\sqrt{-\sigma_R}) + \frac{2}{t\sqrt{\sigma_{R+1}}} J_1(t\sqrt{\sigma_{R+1}}) \tag{G.105}$$

$$= \frac{\|\mathbf{L}(\mathbf{E})\|}{\lambda_{R+1}(\mathbf{L}(\mathbf{P}^*))} \cdot \frac{2}{t\sqrt{-\alpha\lambda_R(\mathbf{L}(\mathbf{P})) + \frac{1}{n-1}}} I_1\left(t\sqrt{-\alpha\lambda_R(\mathbf{L}(\mathbf{P})) + \frac{1}{n-1}}\right) \tag{G.106}$$

$$+ \frac{2}{t\sqrt{\alpha\lambda_{R+1}(\mathbf{L}(\mathbf{P})) - \frac{1}{n-1}}} J_1\left(t\sqrt{\alpha\lambda_{R+1}(\mathbf{L}(\mathbf{P})) - \frac{1}{n-1}}\right). \tag{G.107}$$

Whenever

$$\|\mathbf{L}(\mathbf{E})\| \ll \lambda_{R+1}(\mathbf{L}(\mathbf{P}^*)), t^2\left(\alpha\lambda_R(\mathbf{L}(\mathbf{P})) - \frac{1}{n-1}\right) \to 0 \tag{G.108}$$

with $\alpha\lambda_{R+1}(\mathbf{L}(\mathbf{P})) \gg \frac{1}{n}, t^2\alpha\lambda_{R+1}(\mathbf{L}(\mathbf{P})) \to \infty$, we have

$$\lim_{(t,n)\to\infty} \frac{\|\mathbf{U}_0\mathbf{U}_0^\top\mathbf{Y}_l(t) - \mathbf{Y}_l(t)\|_2}{\|\mathbf{Y}_l(0)\|_2} = 0. \tag{G.109}$$

Also,

$$\frac{\|\mathbf{U}_0\mathbf{U}_0^\top\mathbf{Y}_l(t) - \mathbf{U}_0\mathbf{U}_0^\top\mathbf{Y}_l(0)\|_2}{\|\mathbf{Y}_l(0)\|_2} \tag{G.110}$$

$$= \frac{\|\mathbf{U}_0\mathbf{U}_0^\top\mathbf{U}\boldsymbol{\Gamma}_1(t)\mathbf{y}_{l,1:R}^{(0)} + \mathbf{U}_0\mathbf{U}_0^\top\mathbf{U}_\perp\boldsymbol{\Gamma}_2(t)\mathbf{y}_{l,R+1,n}^{(0)} - \mathbf{U}_0\mathbf{U}_0^\top\mathbf{U}\boldsymbol{\Gamma}_1(0)\mathbf{y}_{l,1:R}^{(0)} - \mathbf{U}_0\mathbf{U}_0^\top\mathbf{U}_\perp\boldsymbol{\Gamma}_1(0)\mathbf{y}_{l,R+1:n}^{(0)}\|_2}{\|\mathbf{Y}_l(0)\|_2} \tag{G.111}$$

$$= \frac{\|\mathbf{U}_0\mathbf{U}_0^\top\mathbf{U}(\boldsymbol{\Gamma}_1(t) - \boldsymbol{\Gamma}_1(0))\mathbf{y}_{l,1:R}^{(0)}\|_2}{\|\mathbf{Y}_l(0)\|_2} + \frac{\|\mathbf{U}_0\mathbf{U}_0^\top\mathbf{U}_\perp(\boldsymbol{\Gamma}_2(t) - \boldsymbol{\Gamma}_2(0))\mathbf{y}_{l,R+1:n}^{(0)}\|_2}{\|\mathbf{Y}_l(0)\|_2} \tag{G.112}$$

$$\leq \|\boldsymbol{\Gamma}_1(t) - \boldsymbol{\Gamma}_1(0)\|_2 + \|\mathbf{U}_0^\top\mathbf{U}_\perp(\boldsymbol{\Gamma}_2(t) - \boldsymbol{\Gamma}_2(0))\|_2 \tag{G.113}$$

$$\leq \left|\frac{2}{t\sqrt{-\sigma_R}}I_1(t\sqrt{-\sigma_R}) - 1\right| + \frac{\|\mathbf{L}(\mathbf{E})\|}{\lambda_{R+1}(\mathbf{L}(\mathbf{P}^*))}\cdot\left|\frac{2}{t\sqrt{\sigma_{R+1}}}J_1(t\sqrt{\sigma_{R+1}}) - 1\right| \tag{G.114}$$

$$= \left|\frac{2}{t\sqrt{-\alpha\lambda_R(\mathbf{L}(\mathbf{P})) + \frac{1}{n-1}}}I_1\left(t\sqrt{-\alpha\lambda_R(\mathbf{L}(\mathbf{P})) + \frac{1}{n-1}}\right) - 1\right| \tag{G.115}$$

$$+ \frac{\|\mathbf{L}(\mathbf{E})\|}{\lambda_{R+1}(\mathbf{L}(\mathbf{P}^*))}\cdot\left|\frac{2}{t\sqrt{\alpha\lambda_{R+1}(\mathbf{L}(\mathbf{P})) - \frac{1}{n-1}}}J_1\left(t\sqrt{\alpha\lambda_{R+1}(\mathbf{L}(\mathbf{P})) - \frac{1}{n-1}}\right)\right|. \tag{G.116}$$

Whenever the condition (G.108) holds, we also have

$$\lim_{(t,n)\to\infty} \frac{\|\mathbf{U}_0\mathbf{U}_0^\top\mathbf{Y}_l(t) - \mathbf{U}_0\mathbf{U}_0^\top\mathbf{Y}_l(0)\|_2}{\|\mathbf{Y}_l(0)\|_2} = 0. \tag{G.117}$$

Eq. (G.108) is ensured by the conditions of the theorem and the asymptotic behavior in (F.71), (G.95) as follows:

The condition $\|\mathbf{L}(\mathbf{E})\| \ll \lambda_{R+1}(\mathbf{L}(\mathbf{P}^*))$ comes from $\lambda_{R+1}(\mathbf{L}(\mathbf{P}^*)) \gg \max\{(t^2\alpha)^{-1}, \|\mathbf{L}(\mathbf{P}^* - \mathbf{P})\|\}$. $t^2\left(\alpha\lambda_R(\mathbf{L}(\mathbf{P})) - \frac{1}{n-1}\right) \to 0$ comes from $t^2\|\mathbf{L}(\mathbf{P}^* - \mathbf{P})\| = o(1)$ as $n \to \infty$ in (T2.N) and $t = o(n^{1/2})$ in (T1.N). Also, with the Weyl's inequality $|\lambda_i(\mathbf{L}(\mathbf{P})) - \lambda_i(\mathbf{L}(\mathbf{P}^*))| \leq \|\mathbf{L}(\mathbf{E})\|$, $\alpha\lambda_{R+1}(\mathbf{L}(\mathbf{P})) \geq \alpha\mathbf{L}(\mathbf{P}^*) - \alpha\|\mathbf{L}(\mathbf{E})\| \gg \frac{1}{n}$ due to $\alpha \gg [n\lambda_{R+1}(\mathbf{L}(\mathbf{P}^*))]$ in (T1.N). Finally, $t^2\alpha\lambda_{R+1}(\mathbf{L}(\mathbf{P})) \to \infty$ can be deduced with Weyl's inequality and $\lambda_{R+1}(\mathbf{L}(\mathbf{P}^*)) \gg (t^2\alpha)^{-1}$ in (T2.N).

# H  Lyapunov exponents for each eigensolution

The expressions (16) or (24) can be interpreted as the temporal evolution of a dynamical system. By examining the Lyapunov exponents, one can gain insight into the asymptotic behaviors for $t \gg 1$.

## H.1  Lyapunov exponent for GD and MM

We calculate the Lyapunov exponent for MM[6]. We pick up the $i^{th}$-term from the equation (16), and set Lyapunov exponent as $\lambda_{Lyap,i}$ for $i \in [n]$:

$$\lambda_{Lyap,i} = \lim_{t\to+\infty}\lim_{\delta\mathbf{Y}_{l,i}(0)\to 0}\frac{1}{t}\log\frac{\|\delta\mathbf{Y}_{l,i}(t)\|_2}{\|\delta\mathbf{Y}_{l,i}(0)\|_2}, \tag{H.118}$$

---

[6]For GD, consistently interpret $m = 0$ throughout this subsection.

where $\delta \mathbf{Y}_{l,i}(t)$ the perturbation at time $t$ caused by the initial perturbation $\delta \mathbf{Y}_{l,i}(0)$. Now, we have

$$\mathbf{Y}_{l,i}(t) = \exp\left(-\frac{t}{1-m}\left(\alpha\lambda_i - \frac{1}{n-1}\right)\right)(\mathbf{u}_i^\top \mathbf{y}_l^{(0)})\mathbf{u}_l, l \in [2]. \tag{H.119}$$

Therefore, by canceling out the same terms in the numerator and the denominator,

$$\lambda_{Lyap,i} = \lim_{t \to +\infty} \lim_{\delta Y_{l,i}(0) \to 0} \frac{1}{t} \log \exp\left(-\frac{t}{1-m}\left(\alpha\lambda_i - \frac{1}{n-1}\right)\right) \tag{H.120}$$

$$= -\frac{1}{1-m}\left(\alpha\lambda_i - \frac{1}{n-1}\right). \tag{H.121}$$

Note that the result holds regardless of $l \in [2]$. With the Lemma 5.5, we have the following proposition

**Proposition H.1.** *(Lyapunov exponent for GD and MM) Under the conditions (T1.M), (T2.M), $n \gg 1$ and $i \in [n]$,*

$$\lambda_{Lyap,i} = -\frac{1}{1-m}\left(\alpha\lambda_i - \frac{1}{n-1}\right). \tag{H.122}$$

*Especially, $\lambda_{Lyap,i} \geq 0$ ($i = 1, \ldots, R$), and $\lambda_{Lyap,i} < 0$ ($i = R+1, \ldots, n$).*

This statement suggests that the elements of $\mathbf{Y}_l(t)$ ($i = 1, \ldots, R$) highlight a specific structure within the cluster, enhancing its distinctive characteristics.

### H.2  Lyapunov exponent for NAG

Now, for the case of NAG. Using the formula (24), the asymptotic approximation (G.95) and (G.96), we arrive at the following conclusion:

**Proposition H.2.** *(Lyapunov exponent for NAG) Under the conditions (T1.N), (T2.N) and $n \gg 1$, we have*

$$\lambda_{Lyap,i} = \begin{cases} \sqrt{-\alpha\lambda_i + \dfrac{1}{n-1}}, & (i = 1, \ldots, R). \\ 0, & (i = R+1, \ldots, n). \end{cases} \tag{H.123}$$

*Especially, $\lambda_{Lyap,i} \geq 0 (i = 1, \ldots, R)$.*

It also suggests that the elements of $\mathbf{Y}_l(t)$ ($i = 1, \ldots, R$) emphasize a cluster structure.

The case for $i = R+1, \ldots, n$ differs slightly. In GD or MM, $\lambda_{Lyap,i}$ is negative, causing to exponential decay. However, in NAG, oscillations occurs, so the Lyapunov exponent is considered to be 0. This aligns with the typical differences between GD, MM and NAG.

## I  Supplemental numerical experiments

### I.1  Differences between solution paths of ODE corresponding to GD, MM and NAG

We derived ODEs corresponding to the iterative optimization procedures for t-SNE with and without acceleration. We also derived their closed-form solution. We compare the difference between solution paths by simply plotting them in Fig. 4. As illustrated in Fig.4a, for $i \in [R]$, i.e., components corresponding to the negative eigenvalues of the Laplacian $\mathbf{L}(\alpha\mathbf{P} - \mathbf{H}_n)$ (let's denote $\sigma < 0$), GD follows an exponential function $\exp(-t\sigma)$, and the MM follows $\exp(-\frac{\sigma t}{1-m})$, both of which are exponential functions (see Eq. (16)). On the other hand, NAG follows a function $\frac{2I_1(t\sqrt{-\sigma})}{t\sqrt{-\sigma}}$ involving the modified Bessel function of the first kind $I_1(\cdot)$ (see Eq. (24)). Based on their series expansions, these functions are greater than 1 for positive values of $t$ and converge to 1 as $t$ approaches zero (see Lemma F.1 for NAG case).

By contrast, in Fig. 4b, for $t \gg 1$, GD and MM decay exponentially and NAG decays under the formula (G.95). From these conditions, the eigenvectors $\mathbf{u}_i$ for $i \geq R+1$, i.e., components corresponding to positive eigenvalues, asymptotically decay to 0 as time $t$ increases.

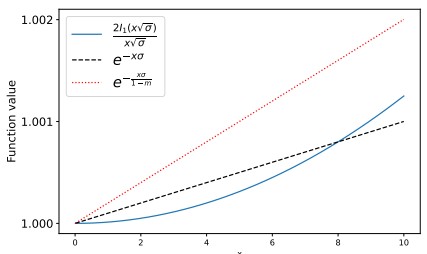
(a) GD, MM and NAG (negative eigenvalues)

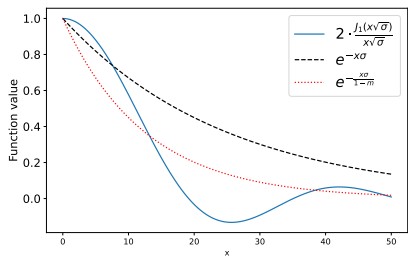
(b) GD, MM and NAG (positive eigenvalues)

Figure 4: Comparison of (Modified) Bessel function and exponential functions.

## I.2 Experiments with synthetic dataset

We conduct a comparison between iterative optimization methods for t-SNE and ODEs corresponding to them, specifically for GD, MM, and NAG, to explore their convergence behaviors using a synthetic dataset, generated from the following Gaussian Mixture Model (GMM):

$$X_i | z_i = r \sim N(\boldsymbol{\mu}_r, \Sigma), \quad z_i \sim \text{Multinomial}(\pi_1, \pi_2, \pi_3), \text{ for } i \in [200], \tag{I.124}$$

where $\boldsymbol{\mu}_1 = (0, 0, 0)^\top, \boldsymbol{\mu}_2 = (150, -110, 170)^\top, \boldsymbol{\mu}_3 = (-130, 150, -150)^\top$ are the centers of Gaussian distributions, $\Sigma = \begin{pmatrix} 30 & 20 & 25 \\ 20 & 50 & 10 \\ 25 & 10 & 30 \end{pmatrix}$ is the common covariance matrix and $(\pi_1, \pi_2, \pi_3) = (1/3, 1/3, 1/3)$. With the above $\{z_i\}_{i \in [200]}$, we define the element of $\mathbf{P}^*$ as $p_{ij}^* = p_{ij}$ if $z_i = z_j$; otherwise, it is set to 0. The Laplacian $\mathbf{L}(\alpha \mathbf{P} - \mathbf{H}_n)$ has an eigenvalue with almost 0 and other two negative eigenvalues, i.e. $R = 3$. According to Cai & Ma (2022), it can be proven that the conditions (T1.M), (T2.M), (T1.N) and (T2.N) are satisfied by placing additional assumptions on these.

Figure 5 shows that the comparison between iterative optimization methods with and without acceleration (GD, MM, and NAG), and ODEs correspond to them for the dataset generated from the above explained GMM. The optimization is done with step size parameter $h = 5$, momentum parameter $m = 0.5$, Perplexity is 30, and exaggeration parameter $\alpha = 10$. In the plot, solid lines indicate iterative approaches and dashed lines denote methods using ODEs. From this plot, we see that the KL-divergence (the objective function of the t-SNE optimization procedure) aligns well between iterative algorithms and continuous limit ODEs. It can also be observed that as $t$ increases, the discrepancy in the KL-divergence values between those obtained from the iterative algorithm and those obtained as the solution path of the ODE becomes larger. This is consistent with the theoretical results obtained in this paper.

As in section 8, we state the assumption in Lemma 5.1 about GD and MM, we identify the variables as $t = kh$, where the variable $t$ as time parameter, $k$ as iteration number and $h$ as step size. For NAG, the identification can be $t = k\sqrt{h}$ as in Lemma 6.1. Figure 6 shows the representation where the top row corresponds to GD, the middle row to MM, and the bottom to NAG.

We have two kinds of observations.

- Case 1: Focus on that the middle case in GD ($t = 35$) and the rightmost case in NAG ($t = 33.54$). In the case of NAG, even though the value of $t$ is smaller at 33.54 compared to 35 in the case of GD, the data is well-separated, whereas in GD it is still undifferentiated.

- Case 2: The rightmost case in GD ($t = 75$) and the middle case in MM ($t = 35$). Formally, the time parameter $t$ is different, and the low-dimensional representation looks similar. Considering the effect of the momentum term $m = 0.5$, it can be interpreted as accelerating time $t$, effectively making it $t = 70 (= 35/(1 - 0.5))$. See the formula (16), where we adopt that MM can be understood as fast-forwarding GD with respect to the variable $t$ thanks to the moment term with $m$.

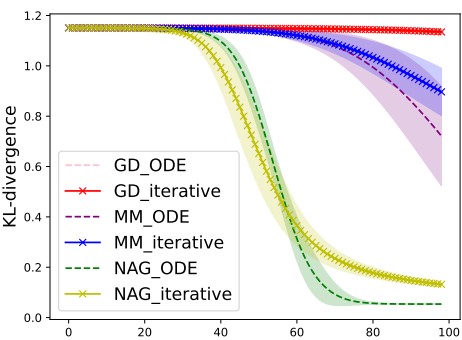

Figure 5: A comparison between iterative optimization methods and those approximated using ODEs (GD, MM, NAG).

### I.3   Profiles of datasets

In addition of GMM datasets explained in section I.2, the following datasets are used. Here are the links and their licenses:

- KDDCup1999: `https://kdd.ics.uci.edu/databases/kddcup99/kddcup99.html`, CC BY 4.0 license referenced by `https://archive.ics.uci.edu/dataset/130/kdd+cup+1999+data`

- MNIST:   `https://scikit-learn.org/stable/modules/generated/sklearn.datasets.fetch_openml.html`, BSD license provided by `sklearn`

- Olivetti Faces:   `https://scikit-learn.org/stable/modules/generated/sklearn.datasets.fetch_olivetti_faces.html`, BSD license provided by `sklearn`

For the KDDCup1999 dataset, download the *kddcup.data_10_percent.gz* file locally and use it. For the MNIST and Olivetti Faces datasets, we download and use them directly in the code by utilizing the functions provided by `sklearn`.

### I.4   Experiments through variation of random Initialization

It is well-known that the clustering results using t-SNE may vary with variations of initialization such as Kobak & Linderman (2021). Cai & Ma (2022) states a sufficient condition in their Theorem 14 as random initialization, which leads to intercluster repulsion (Th.13) after both early exaggeration and embedding stage. It is reported that false clustering may appear due to an incidental combination of overlapped cluster from the early exaggeration stage as in Remark 16 there. By contrast, we focus only on early exaggeration stage, and numerical experiments on several datasets, performed multiple runs with different initializations for the same dataset, and present the evaluation results.

We use KDDCup1999, MNIST and Olivetti datasets. The Olivetti face data set consists of images of 40 individuals with small variations in viewpoint. The data set consists of 400 images (10 per individual) of size $64 \times 64 = 4096$ and is labeled according to identity.

In order to evaluate the variation of initialization, we used Adjusted Rand Index (ARI) (Hubert & Arabie, 1985), which is commonly employed to evaluate clustering performance with known labels (ground truth clusters). This index assesses the degree of correspondence between the predicted clusters and the true class labels, serving as a measure of clustering accuracy in cases where the true labels are available, akin to supervised learning.

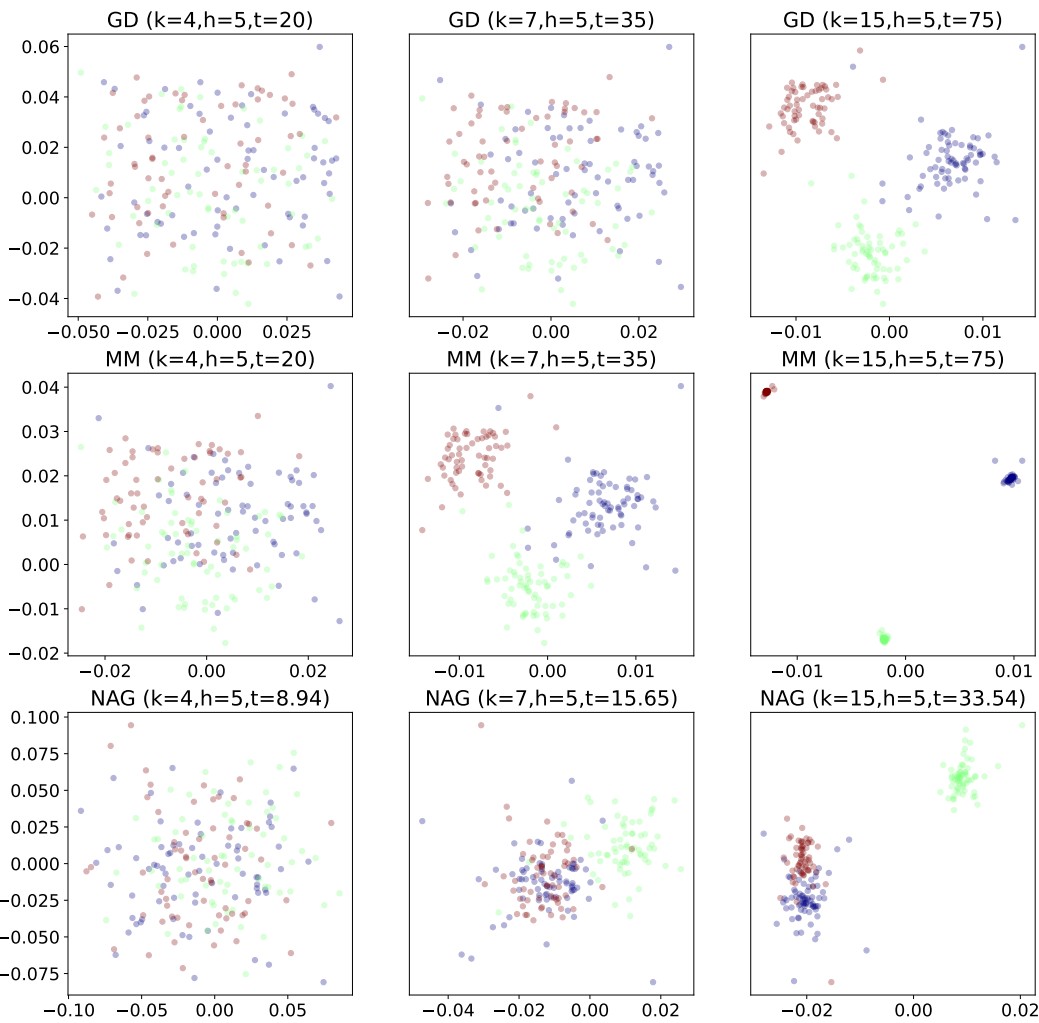

Figure 6: Low-dimensional expression with GMM dataset. Top: GD, Middle: MM, Bottom: NAG.

Figure 7 demonstrates the results. The band visible in the ARI transition represents the results of varying ARR after randomly changing the initial values 30 times. The width of the band corresponds to the standard deviation of the ARR.

In all three datasets, ARI increases most rapidly for NAG(green), followed by MM(purple), with GD(pink) trailing behind. NAG achieves the best results in the shortest time. It is important to note the relationship between the variables $t$ and $k$ for GD and MM, $t = kh$, while for NAG, $t = k\sqrt{h}$. As a result, even with the same value of $k$, the effective value of $t$ is smaller for NAG, which makes the NAG curve appear truncated.

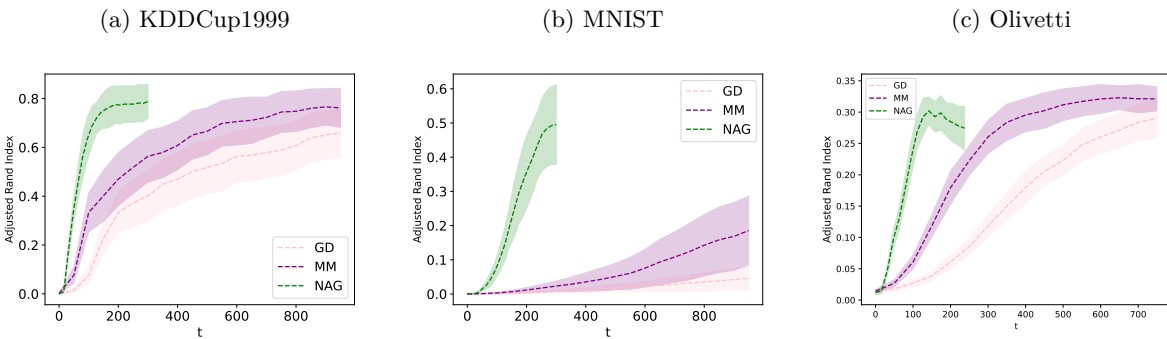

Figure 7: Adjusted Rand Index(ARI) for GD, MM, NAG across different datasets

## I.5 Experiments through various initialization methods beyond random initialization

Generally, there are several ways to initializing methods for t-SNE. Kobak & Linderman (2021) reports the result with initialization using PCA, Linderman & Steinerberger (2019) uses spectral clustering in initializing. In this section, we experiment several initialization methods and compare the results under the equation (16) or (24) with the indicator ARI.

We basically follow the way of section I.4. The MNIST dataset is used for the experiments, and the Adjusted Rand Index (ARI) is employed as the evaluation metric. To assess the stability of each initialization method, we conduct 10 trials for each approach. In addition to random initialization as described in section I.4, we attempt representative initializations using PCA, Spectral Embedding (SE) (Belkin & Niyogi, 2003), and MultiDimensional Scaling (MDS) (Torgerson, 1952). For PCA, SE, and MDS, the dimension is reduced to two by setting `n_components` to 2. In the case of SE, the number of neighbors (`n_neighbors`) used to construct the Laplacian matrix is set to 300.

Figure 8 demonstrate the results. For the optimization methods GD(orange) and MM(purple), we observe that changing the initialization from random to other methods results in starting with a relatively higher ARI, and the metric generally improved as the time $t$ increase.

On the other hand, for NAG (green), it is observed that the ARI starts near zero in all initialization methods. This is because, in the case of GD and MM, all components of the initial values are included in the eigendecomposition of Eq. (16) like $\mathbf{u}_i^\top \mathbf{y}_l^{(0)}$, whereas for NAG, the decomposition in Eq. (24) only includes a subset of the initial values in each term like $\mathbf{U}\boldsymbol{\Gamma}_1(t)\mathbf{y}_{l,1:R}^{(0)}$ or $\mathbf{U}_\perp\boldsymbol{\Gamma}_2(t)\mathbf{y}_{l,R+1:n}^{(0)}$. [7] In the case of NAG, as time progresses, the influence of the eigenvectors becomes reflected, and the ARI gradually improves.

Furthermore, across the different initialization methods, it is confirmed that the SE leads to the greatest improvement in terms of ARI in our experiment.

As a minor point, for MDS, using random placement during initialization results in a similar variation in ARI outcomes as observed with random initialization. In contrast, PCA and SE, being deterministic methods, do not exhibit such variation.

## I.6 Experiments through ARR for other data sets

We also explore the stopping time to obtain the clustering results with ARR(Average Residual Rate) defined in (28) at appropriate time. In addition, we observe whether another indicator Trustworthiness exhibits a similar trend to ARR during the optimization process.

In section 8.2, we evaluate ARR for MNIST data set. Here, we evaluate it with KDDCup1999, Olivetti Face datasets as well. Figure 9 shows the results for KDDCup1999, and Fig. 10 for Olivetti.

---

[7]Remember that both $\boldsymbol{\Gamma}_1(t)$ and $\boldsymbol{\Gamma}_2(t)$ are diagonal matrices as in Theorem 6.4.

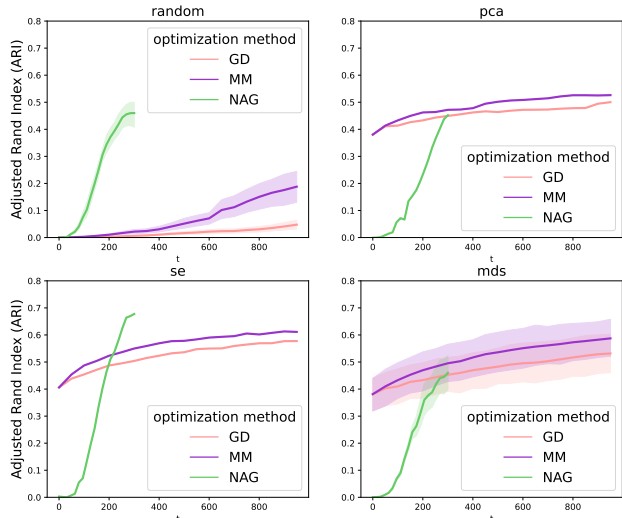

Figure 8: A comparison among various initialization methods with MNIST dataset for GD/MM/NAG

The left figure shows the transition of ARR and Trustworthiness. We begin by reviewing ARR. NAG(green) exhibits the fastest drop in ARR, followed by MM(orange) and then GD(blue). A red dashed line marks the threshold at $ARR = 0.01$. As mentioned in section I.4, it is important to note that due to the different relationships between the variables $t$ and $k$ among GD/MM/NAG[8], the NAG curve appears truncated.

Next, we compare ARR with Trustworthiness (Venna & Kaski, 2001). Trustworthiness is a standard metric used in methods such as t-SNE to evaluate how well the structure of the original high-dimensional data is preserved in the low-dimensional embedding. Specifically, it measures whether the neighbors in the embedded space are also neighbors in the original high-dimensional space. The value ranges from 0 to 1, with values closer to 1 indicating higher agreement.

The same figure also shows the case where the number of neighbors is set to 10 for Trustworthiness. The dashed blue/orange/green lines represent the Trustworthiness values for each optimization method: GD/MM/NAG. As ARR decreases, Trustworthiness tends to increase, indicating an approximate inverse correlation. The correlation coefficient is around $-0.85$ for KDDCup1999 and $-0.90$ for Olivetti. Furthermore, because the Trustworthiness value is sensitive to the choice of neighborhood size, the strong inverse correlation observed in our results does not necessarily indicate that trustworthiness can be used as a general alternative to ARR for determining the stopping time.

Although Continuity is also a well-known metric (Kaski et al., 2003) similar to Trustworthiness, numerical experiments in this case showed that its values were nearly equivalent—though not identical—to those of Trustworthiness. Therefore, we only present Trustworthiness in the figure.

The right panels of figures 9 and 10 show the clustering results of GD, MM, and NAG when $ARR = 0.01$.

### I.7 Experiments with various momentum coefficients

We explore the effect of the momentum coefficient $m$ in the MM. In the Eq. (16), the transition of ARR for MNIST dataset when changing the coefficient m from 0.1 to 0.9 can be seen in Fig. 11. Other configurations of MNIST are the same as in section 8.2. The coefficient $m$ was originally assumed to range from 0 to 1, and it can be confirmed that as $m$ increases, the influence of the residual term on ARR diminishes over a shorter period of time. Formally, GD can be regarded as having $m$ of 0, which is also consistent with the results shown with both datasets -KDDCup1999 and Olivetti- in Figs. 9, 10.

---

[8]$t = kh$ for GD and MM, whereas $t = k\sqrt{h}$ for NAG.

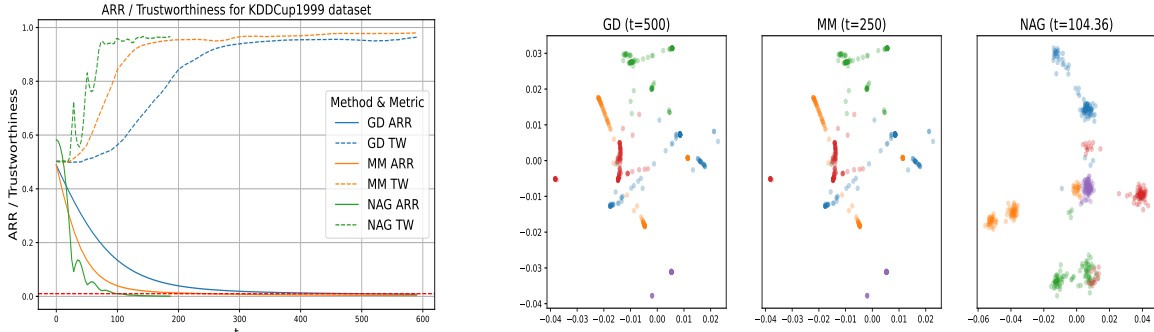

Figure 9: (left): transition of ARR and Trustworthiness (TW), (right): Clustering result for KDDCup1999 dataset with ARR=0.01 using three ODEs: GD, MM, and NAG.

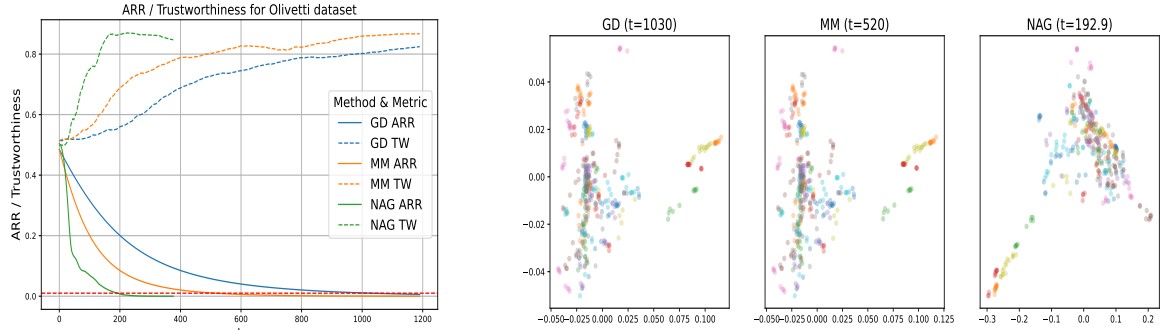

Figure 10: (left): transition of ARR and Trustworthiness (TW), (right): Clustering result for Olivetti Face dataset with ARR=0.01 using three ODEs: GD, MM, and NAG.

### I.8 Further analysis of KDDCup1999 dataset

We obtain the adjacency matrix $\mathbf{P}$ as depicted in Fig. 12. The heatmap illustrates that the elements of $\mathbf{P}$ exhibit relatively high values (orange or yellow) between clusters, while other regions are closer to black.

Solving the eigenproblem with the Laplacian matrix $\mathbf{L}(\alpha\mathbf{P} - \mathbf{H}_n)$ using $\alpha = 10$, we obtain eigenvalues close to zero and four negative eigenvalues, as shown in Fig. 13. They are emphasized during clustering regardless of the method used (GD/MM/NAG), resulting in the formation of clusters in the low-dimensional representation.

Furthermore, Fig.14 displays the distribution of the dominant eigenvectors $\mathbf{u}_1, \ldots, \mathbf{u}_5$. These eigenvectors are emphasized in the low-dimensional representation as the time parameter $t$ progresses under Eqs.(16) and (24). As shown in this figure, the components belonging to the same cluster simultaneously take on the same values, and the relative magnitudes of these values become crucial when forming clusters in the low-dimensional space. This mechanism leads to the clustering results depicted in Fig. 1 as the low-dimensional representation.

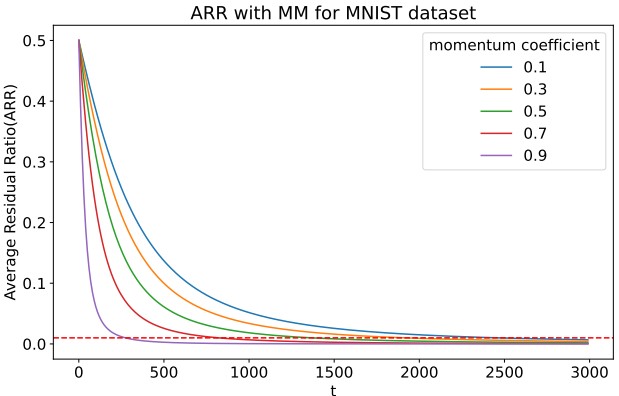

Figure 11: A comparison of MM with various momentum coefficients

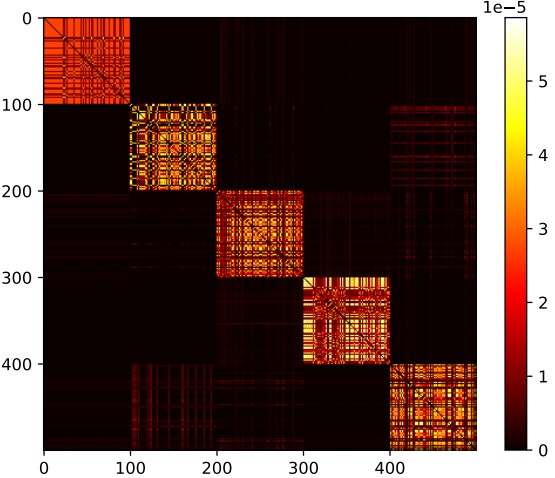

Figure 12: A heatmap of similarity matrix $\mathbf{P}$ for the $n = 500$ samples from KDD Cup 1999 dataset.

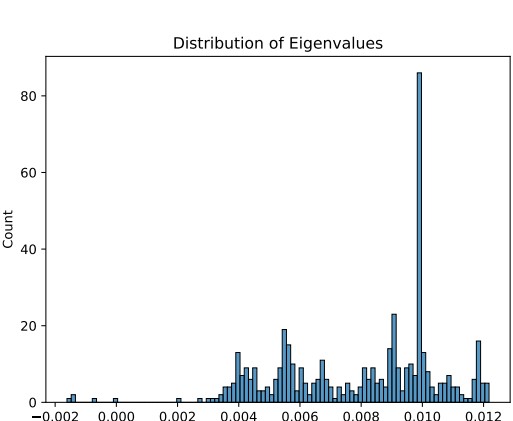

Figure 13: Distribution of eigenvalues $\{\sigma_i\}_{i \in [n]}$ of the Laplacian matrix $\mathbf{L}(\alpha\mathbf{P} - \mathbf{H}_n)$: $\sigma_1 \approx 0$, $\sigma_i < 0 (2 \leq i \leq 5)$ and $\sigma_i > 0 (i \geq 6)$.

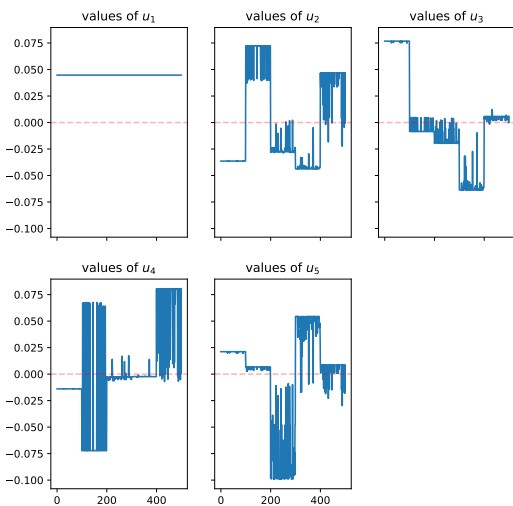

Figure 14: The eigenvectors, whose eigenvalues are the smallest five: $\mathbf{u}_1, \ldots, \mathbf{u}_5$.

