# OpenReview forum: "Dynamics of the accelerated t-SNE"
_TMLR — Accepted by TMLR_

### Review · Reviewer_cTTp · 2025-04-19

**Summary Of Contributions:**

t-SNE is probably the most popular non-linear dimensionality reduction method within data science for visualizing high-dimensional data in 2 dimensions.  Despite this popularity, it is not understood what formally can be concluded about its resulting embeddings, and how to understand them.  This paper improves this understanding by studying its iterative dynamics -- in particular, how it is affected by using momentum or Nesterov's accelerated gradient.

The main results show that (and how, when) these optimization improvements help -- in general they lead to faster convergence.  It also exposes (through a closed for solution of an ODE of the system) a time parameter, and how the resulting embeddings strongly depends on it.

**Audience:**

Yes

**Claims And Evidence:**

Yes

**Requested Changes:**

Suggestions:
 I think the potential impact of the paper could be *significantly* improved if the authors add a subsection (after "main contributions" before Sec 2), about something like "Take-a-ways for Users of t-SNE".  I am imagine something like:
  - if t-SNE is too slow for your data set (and some easy to understand condition holds) run with MM  (or NAG)
  - The time parameter (explain in very simple terms, and how one can tune it in software) plays an important role, similar to the bandwidth choice.  Set it large for tighter clusters [or some similar broad guidance].

I trust the authors to improve upon my sketched ideas above.

**Strengths And Weaknesses:**

Strengths:
  + t-SNE is a highly, highly used procedure, and its full understanding as eluded the community (for the most part) since its introduction, so work on this topic is important
  + The analysis is mathematically rigorous, involves blending of understanding from several domains, and ties it together in a careful way
  + The main mathematical insights are supported and enhanced by simple experiments
  + The use of momentum and NAG are demonstrated to improve the efficiency, and explained why and when.


Weaknesses:
  + The results are explained in a highly technical way that will mostly only be accessible to experts in both t-SNE and optimization.  While this is an appropriate framing for others who would investigate these issues, it will have less impact on the much, much broader community that simply use t-SNE (mostly as a black box).
  + Much of the work is an extension of the work of Cai & Ma (JMLR 2022).

---

> ### Author Response · Authors · 2025-05-07
>
> > The results are explained in a highly technical way that will mostly only be accessible to experts in both t-SNE and optimization. While this is an appropriate framing for others who would investigate these issues, it will have less impact on the much, much broader community that simply use t-SNE (mostly as a black box).
>
> We agree that the presentation is technical and primarily intended for readers with a background in t-SNE and optimization.
> Our main goal in this work is to provide a deeper theoretical understanding of how momentum-based methods affect the dynamics of t-SNE, which we believe is foundational for future developments and refinements of dimensionality reduction techniques.
>
> While this may limit immediate accessibility for users who treat t-SNE as a black box, we hope that clarifying these underlying mechanisms will ultimately inform better algorithmic designs and practical implementations that can benefit the broader community.
>
> We appreciate the reviewer’s comment and have added a brief note in the introduction to clarify for more broad audience.
>
> > Much of the work is an extension of the work of Cai & Ma (JMLR 2022).
>
> We believe that extending the analysis from GD to momentum-based methods, particularly NAG, is non-trivial.
> The introduction of second-order dynamics leads to qualitatively different behavior, including Bessel and modified Bessel dynamics, which significantly alter the clustering process during the early exaggeration phase.
>
> Furthermore, we propose the Average Residual Ratio (ARR) as a new metric to diagnose the clustering progression based on spectral properties, which is not present in previous work.
>
> Thus, while our work is built on existing foundations, we aim to provide novel theoretical insights into accelerated t-SNE dynamics that go beyond the original GD-based framework.
>
> > Suggestions:
> > I think the potential impact of the paper could be significantly improved if the authors add a subsection (after "main contributions" before Sec 2), about something like "Take-a-ways for Users of t-SNE". I am imagine something like:
> > - if t-SNE is too slow for your data set (and some easy to understand condition holds) run with MM (or NAG)
> > - The time parameter (explain in very simple terms, and how one can tune it in software) plays an important role, similar to the bandwidth choice. Set it large for tighter clusters [or some similar broad guidance].
> > I trust the authors to improve upon my sketched ideas above.
>
> Following your advice, we have added a new subsection titled ""1.2 Key Takeaways for t-SNE Users"" after the main contributions section and before Section 2.
>
> This subsection provides a brief and accessible summary for practitioners, including practical recommendations such as when to consider using MM or NAG, and guidance on the stop timing with ARR.
>
> We appreciate the reviewer’s helpful sketch and incorporated similar ideas while tailoring them to the context of our analysis.

---

> > ### Comment · Reviewer_cTTp · 2025-05-07
> >
> > Thanks for the update to the paper.  It has satisfied all of my concerns, and I think made the paper more useful.

---

### Review · Reviewer_8vBG · 2025-04-21

**Summary Of Contributions:**

The paper analyzes the t-stochastic neighbor embedding (t-SNE) mwrhos. The key contribution is the analysis of t-SNE with included momentum term from the dynamical systems perspective. The analysis shows better convergence rates of the variant with momentum term. The authors further extend to applying Nesterov's accelerated gradient (NAG) method to t-SNE to improve convergence speed further.

**Audience:**

Yes

**Broader Impact Concerns:**

No concerns.

**Claims And Evidence:**

Yes

**Requested Changes:**

1) In the literature review section, it would be good to spell out how you advance the state-of-the-art. Particularly, in Sec. 2.4, how do you solve the problem you identify?
2) The writing is very dense and focuses on the technical concepts. It would be great to provide some intuition and explain what the different concepts mean. What is the impact of assumptions. What is the impact of the results you derive.
3) Please make sure all elements are properly introduced and defined. For instance, in the beginning of Sec. 4, I don't think H_n was introduced. Also, why is the reformulation for which you use H_n important?
4) Can you add a concrete problem formulation?
5) Can you clarify whether Theorem 4.1 is taken from the reference (Cai & Ma, 2022) or has some novel elements?
6) The concept of a stopping time has a specific definition in stochastic analysis. But to my understanding, you are using it more as a general criterion for when to stop an algorithm. I think it would be good to make that clear.
7) The language should be improved. There are some plural 's' missing, e.g., in the first and second sentence of the second paragraph in the introduction, some articles are missing, e.g., when defining the adjacency matrix after (4), "where put m = 0 for GD" sounds a bit off.

**Strengths And Weaknesses:**

Strengths:
1) The authors provide a theoretical analysis of a widely used method.
2) The closed-form ODE solutions should be of practical benefit.

Weaknesses:
1) I found the paper very difficult to follow. This is for one certainly because I am not familiar with t-SNE. But it is also due to the writing, which is very dense, does not provide much intuition, and is missing some definitions. Also the language should be improved.
2) The paper is missing a problem formulation.
3) It is not always exactly clear what is taken from related literature and what is a novel contribution.

---

> ### Author Response · Authors · 2025-05-07
>
> >I found the paper very difficult to follow...
>
> We appreciate your honesty and fully understand that the paper was difficult to follow, especially for readers not already familiar with t-SNE. We take this as valuable input on the presentation and accessibility of our work.
> In the revised version, we have taken the following concrete steps to improve clarity:
>   - We added brief background explanations of t-SNE and early exaggeration, including intuitive motivations and notation in introduction and contribution.
>   - Several missing definitions (e.g., H_n) have been added or clarified.
>   - We revised the writing to improve readability and structure, and smoothed out language issues throughout as much as possible.
>
> We hope these changes make the paper more accessible to a wider audience, and thank you again for highlighting this important concern.
>
> > The paper is missing a problem formulation.
>
> We have addressed this point by explicitly adding a Problem Formulation in the middle of Section 3.
>
> > It is not always exactly clear what...
>
> We have revised the Contributions section to more clearly distinguish between elements based on existing literature and our novel contributions. Indeed, we rephrase our contribution, especially not only practical aspects but theoretical achievements.
>
> > In the literature review section, ..
>
> We have revised Section 2.4 to more clearly position our contribution relative to prior work on accelerating t-SNE.
> Specifically, we now distinguish between practical acceleration methods (e.g., Barnes-Hut, n-body approximations) and our focus on theoretical acceleration in the continuous-time dynamics.
> We also clarify that while our current framework (taking continuous limit and considering ODE) does not aim to improve runtime efficiency, it provides analytical insights—such as Bessel-type oscillations and the proposed ARR metric—that could inspire future algorithmic improvements for large-scale applications, and shown that even with NAG, t-SNE reveals cluster structure in the high-dimensional data. With this theoretical gurantee, we can use NAG for t-SNE with confidence to improve the convergence.
>
> We believe these revisions better clarify how our work advances the theoretical understanding of t-SNE optimization and complements existing acceleration techniques.
>
> We have revised Section 2.4 to more clearly position our contribution relative to prior work on accelerating t-SNE.
>
> Specifically, we now distinguish between practical acceleration methods (e.g., Barnes-Hut, n-body approximations) and our focus on theoretical acceleration in the continuous-time dynamics.
> We also clarify that while our current framework (taking continuous limit and considering ODE) does not aim to improve runtime efficiency, it provides analytical insights—such as Bessel-type oscillations and the proposed ARR metric—that could inspire future algorithmic improvements for large-scale applications, and shown that even with NAG, t-SNE reveals cluster structure in the high-dimensional data. With this theoretical gurantee, we can use NAG for t-SNE with confidence to improve the convergence.
>
> We believe these revisions better clarify how our work advances the theoretical understanding of t-SNE optimization and complements existing acceleration techniques.
>
> > The writing is very dense ...
>
> Following your suggestion, we have revised Sections 4 and 5 by adding intuitive explanations before and/or after each theorem, proposition, and lemma.
> These additions clarify the meaning of the results, the roles and impacts of the assumptions, and the broader implications of the derived statements. We hope these improvements make the technical developments more accessible and help readers better understand the underlying ideas and motivations behind our analysis.
>
> > Please make sure all elements are ...
>
> We have addressed this point by explicitly introducing and defining $H_n$ at the beginning of Section 3.
> We have also added a brief explanation of why this reformulation is important: namely, it simplifies the expression of the dynamics and facilitates the subsequent spectral analysis.
> We hope this clarification makes the logical flow of the paper more transparent.
>
> > Can you clarify whether Theorem 4.1 ..
>
> Theorem 4.1 itself evaluates an approximation error bound for the matrix S, whose definition (4) is independent of whether a momentum term is present. While our update rule (5) incorporates a momentum term and thus modifies the evolution of y and q compared to the GD case, the definition of S remains unchanged.
>
> Consequently, Theorem 4.1 asserts that the approximation bound established by Cai & Ma (2022) for the GD case still holds under the momentum-based update dynamics.
> Thus, while the inequality itself is not fundamentally new, it was no shown in (Cai&Ma,2022) and our contribution is to verify that it remains valid even in the presence of momentum, which is non-trivial due to the altered trajectory of the embeddings.

---

> > ### Author Response · Authors · 2025-05-07
> >
> > > The concept of a stopping time has a specific definition in stochastic analysis. But to my understanding, you are using it more as a general criterion for when to stop an algorithm. I think it would be good to make that clear.
> >
> > We have added a note at the beginning of Section 7 to make it explicit that the term "stopping time" in our paper is used informally to refer to a general criterion for when to stop the algorithm, and is not intended in the technical sense used in stochastic analysis.
> >
> > We appreciate the suggestion to prevent potential confusion.
> >
> > > The language should be improved. There are some plural 's' missing, e.g., in the first and second sentence of the second paragraph in the introduction, some articles are missing, e.g., when defining the adjacency matrix after (4), "where put m = 0 for GD" sounds a bit off.
> >
> > We have carefully reviewed the manuscript and corrected the issues you pointed out, including missing plurals and articles.
> > Specifically, we revised the phrase "where put m=0 for GD" to "where m is set to 0 for GD" to improve fluency and grammatical correctness.
> >
> > We appreciate your careful reading and helpful suggestions for improving the clarity of the writing.

---

> > > ### Comment · Reviewer_8vBG · 2025-05-09
> > >
> > > Thanks for addressing my comments; the paper is now easier to follow, which should increase its impact. I still have some minor comments.
> > >
> > > You sometimes refer to specific theorems/propositions/lemmas in the newly added parts. It seems like a couple of spaces went missing while introducing those. It also seems like much of it is written in the past tense, which makes it read a bit more like a report. Especially in the early stages of a paper, I would rather speak of "we investigate" instead of "we investigated."
> > >
> > > I appreciate adding the key takeaways. My recommendation here would be to focus not only on how to use things in practice, but also on how the theoretical results of the paper are helping the practitioner. I would also think about starting the paragraph a bit differently. The way it reads currently, it seems like everything that follows is only relevant in a very limited special case. Instead, one could also write along the lines of "if we have a problem with characteristics xyz, then..."
> > >
> > > Lastly, I would replace "we put" with "we set" when mentioning that variables are assigned certain values.

---

> > > > ### Author Response · Authors · 2025-05-10
> > > >
> > > > >You sometimes refer to specific theorems/propositions/lemmas in the newly added parts. It seems like a couple of spaces went missing while introducing those.
> > > >
> > > > Thank you for pointing that out. I’ve made sure to include appropriate spacing when referring to specific theorems, propositions, or lemmas in the revised parts. However, if there are any remaining instances where spacing appears off (perhaps due to rendering issues), I’d be happy to double-check and correct them.
> > > >
> > > > >It also seems like much of it is written in the past tense, which makes it read a bit more like a report. Especially in the early stages of a paper, I would rather speak of "we investigate" instead of "we investigated."
> > > >
> > > > We appreciate the suggestion. I’ve revised the relevant sections, including the Contributions and other early parts of the paper, to consistently use the present tense (e.g., “we investigate” instead of “we investigated”) in accordance with your feedback.
> > > >
> > > > >I appreciate adding the key takeaways. My recommendation here would be to focus not only on how to use things in practice, but also on how the theoretical results of the paper are helping the practitioner.
> > > >
> > > > Thank you for the helpful comment. I’ve updated the key takeaway section to include a theoretical perspective at the beginning, to better highlight how the theoretical results support practical usage.
> > > >
> > > > >I would also think about starting the paragraph a bit differently. The way it reads currently, it seems like everything that follows is only relevant in a very limited special case. Instead, one could also write along the lines of "if we have a problem with characteristics xyz, then..."
> > > >
> > > > Thank you for the suggestion. I’ve revised the latter part of the key takeaway section to clarify the practical assumptions and to reframe the paragraph in a way that presents the approach as applicable under well-defined conditions, rather than as a special or narrow case.
> > > >
> > > > >Lastly, I would replace "we put" with "we set" when mentioning that variables are assigned certain values.
> > > >
> > > > Thanks for the suggestion. I’ve gone through the manuscript and replaced all instances of “we put” with “we set” where variables are assigned values.

---

> > > > > ### Comment · Reviewer_8vBG · 2025-05-22
> > > > >
> > > > > I think the way the theoretical key takeaways are presented now is not helping much. To understand them, one needs to read almost the entire paper before. The question would be whether it would be possible to provide some intuition. For what class of problems can we get what kind of results? I think that would be a good addition. However, should that not be possible, I would leave that part out and focus on the practical side.

---

> > > > > > ### Author Response · Authors · 2025-05-23
> > > > > >
> > > > > > Thank you for your insightful comment. We agree that the original phrasing was not sufficiently intuitive, and we have revised the text accordingly.
> > > > > > Since the main body of the paper contains the detailed explanation, we focused on rewriting the takeaway section to ensure it conveys the core message clearly on its own. We believe the revised version improves clarity and allows readers to grasp the key point even without referring to the full discussion.

---

### Review · Reviewer_xx4b · 2025-04-24

**Summary Of Contributions:**

This paper presents a theoretically motivated extension of the dynamical systems view of the t-SNE algorithm by incorporating momentum (MM) and Nesterov’s Accelerated Gradient (NAG). The authors derive closed-form ODE solutions, reveal connections to Bessel functions, and propose an early stopping criterion (ARR) based on spectral decomposition. The approach is novel in framing accelerated t-SNE through analytically solvable continuous-time dynamics, and the paper is well-supported by derivations and empirical results.

**Audience:**

Yes

**Broader Impact Concerns:**

I do not foresee any potential major ethical problem.

**Claims And Evidence:**

Yes

**Requested Changes:**

1. What new understanding of t-SNE's clustering behavior does the proposed method bring that is not already accessible via the GD-based analysis?

2. How does the proposed method handle complexity problem?

3. Can you demonstrate a real-world case where MM or NAG-based t-SNE materially outperforms both standard t-SNE and UMAP?

**Strengths And Weaknesses:**

Strengths
1. The paper develops nontrivial connections between gradient-based optimization and continuous dynamical systems involving Bessel and modified Bessel functions. The derivation of the ODEs under MM and NAG is rigorous.
2. The approach allows analytical computation of embeddings at any time 𝑡, which is conceptually elegant.
3.  The proposed Average Residual Ratio (ARR) is a clever and interpretable metric tied to spectral concentration.

Weaknesses
1. The analysis closely parallels to [1], only extending from GD to MM/NAG. While nontrivial technically, is this sufficient for a full TMLR paper? The key observation that acceleration introduces Bessel-type behavior is interesting but lacks broader implications or surprising insight into t-SNE dynamics.
2. The use of eigendecomposition of 𝐿(𝑃) implies $O(N^3)$ complexity, which contradicts the paper's emphasis on acceleration and efficiency. How does this framework apply to realistic high-dimensional datasets like CIFAR or single-cell RNA-seq with 10K–1M samples?
3. All experiments are limited to small datasets (MNIST, KDDCup1999, 1600 samples). These are too simple for demonstrating any practical edge of the proposed method.
4. The entire analysis is confined to the early exaggeration (EE) stage. However, meaningful embeddings often require a post-EE embedding phase. The paper would benefit from analyzing or adapting the embedding phase itself.

[1] T. Tony Cai and Rong Ma. Theoretical foundations of t-SNE for visualizing high-dimensional clustered data.

---

> ### Author Response · Authors · 2025-05-07
>
> > The analysis closely parallels to [1], only extending from GD to MM/NAG. While nontrivial technically, is this sufficient for a full TMLR paper? The key observation that acceleration introduces Bessel-type behavior is interesting but lacks broader implications or surprising insight into t-SNE dynamics.
>
> > [1] T. Tony Cai and Rong Ma. Theoretical foundations of t-SNE for visualizing high-dimensional clustered data.
>
> We appreciate the reviewer’s observation regarding the connection to [1]. While our approach builds upon the GD-based continuous-time formulation introduced in [1], we respectfully suggest that the extension to MM and NAG reveals distinct dynamical behaviors that are not merely incremental but structurally different.
>
> In particular, we show that when the eigenvalues of the Laplacian $L(P)$ are positive, the continuous-time solution under NAG exhibits Bessel-type dynamics, which are characterized by oscillatory yet decaying behavior. This is in sharp contrast to the monotonic exponential decay seen in GD-based formulations.
>
> Importantly, this behavior is not only mathematically elegant, but also reflected in empirical patterns observed in our proposed Average Residual Ratio (ARR). Specifically, in some datasets, ARR exhibits oscillatory decay trends, consistent with the theoretical prediction derived from Bessel functions. See Figure 9. We believe this observation deepens the understanding of how accelerated methods interact with the graph spectrum in the context of t-SNE dynamics.
>
> > The use of eigendecomposition of $L(P)$ implies complexity, which contradicts the paper's emphasis on acceleration and efficiency. How does this framework apply to realistic high-dimensional datasets like CIFAR or single-cell RNA-seq with 10K–1M samples?
>
> It is true that a full eigendecomposition of the Laplacian $L(P)$ is $O(n^3)$ in general, which poses limitations for very large datasets.
> However, we emphasize that this step is used in our framework only for analytical and interpretative purposes, particularly in computing the proposed ARR metric.
>
> That said, we agree that scalability is an important concern. In future work, the use of approximate spectral methods, such as the Lanczos algorithm (e.g., scipy.sparse.linalg.eigsh) or other partial eigensolvers, can be considered. These methods compute only the top-𝑘 eigenpairs and can reduce the cost to approximately $O(n^2)$ for sparse Laplacians.
>
> Our current implementation prioritizes theoretical clarity and interpretability, but we believe that these approximations would make the framework much more scalable and practical for datasets like CIFAR or single-cell RNA-seq in future work.
>
> Also, as updated in Contributions, our approach guarantees the theoretical basis on iterative methods for MM/NAG through continuous relaxation, because iterative ways are very close to continuous relaxation as Proposition 5.2 or 6.2 say.
>
> > All experiments are limited to small datasets (MNIST, KDDCup1999, 1600 samples). These are too simple for demonstrating any practical edge of the proposed method.
>
> While we acknowledge that the datasets used (e.g., MNIST, KDDCup1999) are modest in size, they were intentionally chosen for their clarity and interpretability, which is crucial for our goal: to analyze and understand the dynamical behavior of accelerated t-SNE variants in a rigorous and transparent setting.
>
> Our contribution is not centered around outperforming baselines on large-scale benchmarks, but rather in developing a complete theoretical framework that reveals new mathematical structure (e.g., Bessel-type dynamics) and introduces a novel interpretable metric (ARR). These contributions stand independently of dataset scale.
>
> We believe that theoretical insight and practical scalability are both valuable—but in this work, we have focused on the former. Scaling to larger datasets, while important, would require dedicated engineering optimizations that go beyond the scope of this paper.

---

> > ### Author Response · Authors · 2025-05-07
> >
> > > The entire analysis is confined to the early exaggeration (EE) stage. However, meaningful embeddings often require a post-EE embedding phase. The paper would benefit from analyzing or adapting the embedding phase itself.
> >
> > It is true that our analysis focuses on the EE stage of the t-SNE algorithm. However, this is an intentional design choice, which is motivated by both the original t-SNE paper by Hinton & van der Maaten and by our specific theoretical goals.
> >
> > The paper "Visualizing data using t-SNE." by van der Maaten & Hinton explicitly introduced EE to promote cluster formation early in the optimization, ensuring that nearby points in high-dimensional space are quickly pulled together. Our study builds on this intuition and aims to provide a rigorous dynamical interpretation of this crucial phase, particularly under momentum-based methods.
> >
> > Since the primary goal of EE—to establish initial cluster structure—is achieved within that stage, we preserve the standard post-EE embedding phase from the original algorithm without modification.
> > This allows us to isolate the effects of acceleration within EE without introducing confounding factors in the later phase.
> >
> > We believe this focus on EE is justified, as it is the stage that most strongly influences the qualitative structure of the final embedding. Extending the analysis to the post-EE phase would be interesting, but is orthogonal to the current paper’s primary goal of characterizing accelerated cluster formation.
> >
> > > What new understanding of t-SNE's clustering behavior does the proposed method bring that is not already accessible via the GD-based analysis?
> >
> > We have clarified this point in the revised "Contributions" section of the paper, and summarize here briefly:
> > Our framework shows that clustering behavior in t-SNE can be understood via spectral decomposition of the Laplacian, where each component evolves according to its Lyapunov exponents.
> >
> >   - Theoretical basis for MM/NAG for iterative convergence via continuous relaxation and their eigen solutions.
> >   - Positive-exponent components dominate the long-term solution and determine the clustering structure, with their spatial configuration given by the eigenvectors.
> >   - In contrast, negative-exponent components decay over time and do not contribute to the final embedding.
> >   - Importantly, under NAG, these stable (negative) modes exhibit oscillatory decay, which may affect the transient dynamics and enhance visual separability during early exaggeration.
> >
> > This offers a refined interpretation of how acceleration alters the temporal profile of cluster formation, grounded in dynamical systems theory.
> >
> > > How does the proposed method handle complexity problem?
> >
> > We agree that a full eigendecomposition of the graph Laplacian incurs high computational cost and does not scale directly to large datasets.
> >
> > However, we would like to clarify that our framework is designed primarily for theoretical understanding and analytical insight, rather than immediate scalability. The eigendecomposition is used to analyze the spectral dynamics and to define the ARR metric, but is not part of a runtime optimization loop.
> >
> > In future work, we believe that approximate methods (e.g., partial eigendecomposition via Lanczos algorithm, randomized projections, etc.) could be used to reduce complexity to around $O(n^2)$ or better, especially when the Laplacian is sparse.
> >
> > We have added a brief discussion of the above in "Conclusion".
> >
> > > Can you demonstrate a real-world case where MM or NAG-based t-SNE materially outperforms both standard t-SNE and UMAP?
> >
> > While we agree that comparing against standard t-SNE and UMAP would be valuable from a practical standpoint, we respectfully note that the goal of our work is not empirical performance comparison, but rather to develop a theoretical understanding of how momentum-based dynamics influence clustering in t-SNE.
> >
> > Our contributions are analytical in nature: we provide explicit solutions, reveal Bessel-type behavior, and introduce the ARR metric—all of which aim to shed light on the structure of the optimization dynamics.
> > That said, we recognize that these insights may inform future improvements in embedding quality or interpretability, and we agree that evaluating such potential benefits empirically -- especially on real-world datasets --  would be a valuable direction for future work.

---

> > > ### Comment · Reviewer_xx4b · 2025-05-26
> > > **Response**
> > >
> > > I appreciate the authors' efforts in the revision, which addresses some of my concerns. Here are some additional suggestions for scalability, practical embedding quality, and end-to-end performance.
> > >
> > > First, the analysis remains limited to the early exaggeration stage, and it is unclear how the findings influence the full t-SNE pipeline.
> > >
> > > Second, the reliance on eigendecomposition hinders scalability. While the authors note this is for theoretical analysis, the limitation should be stated more clearly in the main text.
> > >
> > > Thirdly, the ARR metric is interesting but would benefit from comparisons with standard quality metrics to assess its practical value.
> > >
> > > Finally, the experiments are restricted to small datasets. It would help to show even a modest empirical advantage over standard t-SNE to support the claim of acceleration.

---

> > > > ### Author Response · Authors · 2025-05-27
> > > >
> > > > We appreciate reviewer's observation.
> > > >
> > > > > First, the analysis remains limited to the early exaggeration stage, and it is unclear how the findings influence the full t-SNE pipeline.
> > > >
> > > > As stated in our previous response and in the Introduction, our paper intentionally focuses on a theoretical analysis limited to the EE stage. As shown in Proposition 5.2 for the MM and Proposition 6.2 for NAG, the iterative and continuous formulations are closely approximated.
> > > > These formulations are then connected to the subsequent embedding stage, and as demonstrated in Figure 5, the resulting global cluster structures post-EE—remain comparable to those of the original t-SNE.
> > > >
> > > > > Second, the reliance on eigendecomposition hinders scalability. While the authors note this is for theoretical analysis, the limitation should be stated more clearly in the main text.
> > > >
> > > > We put the description the following description in the first bullet point of Contribution:
> > > > `As these claims suggest, our approach explicitly involves eigenvalues and eigenvectors. When the data size is $n$, solving the eigenvalue problem requires computational complexity $O(n^3)$, which is greater than the original $O(n^2)$ complexity. Therefore, in this paper, we limit our numerical experiments to datasets of relatively small volume.`
> > > >
> > > > > Thirdly, the ARR metric is interesting but would benefit from comparisons with standard quality metrics to assess its practical value.
> > > >
> > > > To the best of my knowledge, stopping criteria in the t-SNE algorithm are determined empirically.
> > > > We understand that the concern is about the validity of the ARR metric.
> > > > In addition to the discussion already provided in Section 8.2, we would kindly refer the reviewer to the numerical experiments shown in Figures 9, 10, and 11 in the Appendix, which include both quantitative evaluations and the resulting visualizations.
> > > >
> > > > > Finally, the experiments are restricted to small datasets. It would help to show even a modest empirical advantage over standard t-SNE to support the claim of acceleration.
> > > >
> > > > In response to Reviewer "8vBG", we have added a clarification in Section 1.2 Key Takeaways, emphasizing the theoretical nature of our contribution.
> > > > Due to the significant computational cost on large-scale datasets, we have deliberately limited our experiments to smaller datasets. As mentioned in Section 9 Conclusion, we also outline future directions for reducing the overall computational cost, such as employing the Lanczos algorithm.

---

### Review · Reviewer_qjTq · 2025-04-24

**Summary Of Contributions:**

The paper investigates the dynamics of t-SNE. It advances our understanding of t-SNE using the perspective of viewing iterative optimization as ODEs and investigate the convergence behavior under accelerated gradients/momentum.

**Audience:**

Yes

**Claims And Evidence:**

Yes

**Requested Changes:**

- Consider giving some type of outline or big picture diagram in the appendix or intro that gives a summary of how the components fit together (i.e. what you are trying to show, what tools allow you to show that, and why those tools are appropriate etc)

- Consider adding a small paragraph outlining exactly which technical components are new and what are borrowed from prior work so it is clear to the reader

Minor changes:

1. Since this is an ML audience, use of physics terminology such as ansatz might not be universally understood in the community. Perhaps can add some clarification/definition here to make it easier for the ML reader, and to clarify what exactly the notation $\approx$ here is referring to.

**Strengths And Weaknesses:**

Strengths:

1. the theoretical derivations, motivation and contributions are in depth and non-trivial, although it can benefit from better exposition.
2. The empirical work gives new insights and is relevant to the corresponding theory
3. The topic is relevant, important and potentially of broad interest

Weaknesses.

1. The exposition is dense and lack enough motivation and big picture.
2. Unclear what part of the analysis is novel versus modifications of prior work.

---

> ### Author Response · Authors · 2025-05-07
>
> > The exposition is dense and lack enough motivation and big picture.
>
> To address your comment, we have revised Sections 3 and 4 by adding explanatory sentences before and after each theorem, proposition, and lemma.
> These additions are intended to provide intuitive guidance, clarify the role of each result within the overall analysis, and improve the logical flow of the presentation.
>
> > Unclear what part of the analysis is novel versus modifications of prior work.
>
> To address this, we have revised the Contributions section to explicitly highlight a new aspect of our work:
> we interpret the spectral decomposition of the dynamics in terms of Lyapunov exponents, offering a novel theoretical perspective on the formation of clusters in t-SNE.
> In particular, we clarify how components with positive Lyapunov exponents dominate the clustering structure, and how, in the case of NAG, stable components exhibit non-monotonic, oscillatory decay governed by Bessel functions.
> We hope this revision better delineates the novel elements of our analysis compared to prior work.
>
> > Consider giving some type of outline or big picture diagram in the appendix or intro that gives a summary of how the components fit together (i.e. what you are trying to show, what tools allow you to show that, and why those tools are appropriate etc)
>
> We thought that clarifying the relationships between the theorems, propositions, and lemmas is essential. Therefore, we added links to the relevant propositions in the Contributions section, and also included explanations of the significance and intuition behind the relationships among propositions in Sections 4 and 5.
>
> > Consider adding a small paragraph outlining exactly which technical components are new and what are borrowed from prior work so it is clear to the reader
>
> To address this point, we have updated the Introduction and added an explicit clarification at the end of the Main Contributions section.
>
> We now clearly distinguish which technical components are new (e.g., the extension to MM and NAG, the emergence of Bessel-type dynamics, and the proposal of the ARR criterion) and which aspects are based on prior work (particularly the continuous-time framework for GD dynamics from Cai & Ma, 2022).
>
> We hope this clarification helps readers more easily understand the novelty and positioning of our contributions.
>
> > (Minor changes) Since this is an ML audience, use of physics terminology such as ansatz might not be universally understood in the community. Perhaps can add some clarification/definition here to make it easier for the ML reader, and to clarify what exactly the notation
>  here is referring to.
>
> To make the terminology more accessible to the broader ML audience, we have replaced all instances of "ansatz" with "assumptions" throughout the manuscript.

---

> > ### Comment · Reviewer_qjTq · 2025-05-09
> > **Response to Authors's Revision**
> >
> > I thank the authors for the revision. I am satisfied with the revision and the authors have addressed my comments adequately.

---

### Author Response · Authors · 2025-05-07

Dear Reviewers and Action Editor,

Thank you for your careful reading and thoughtful suggestions. Reading the review comments, we have updated our paper. Added contents are colored red in the updated paper.
Other changes that reflect the Reviewers’ requests will be discussed in individual responses.

Sincerely,
 yours

---

### Decision · Action_Editor_Pf36 · 2025-05-29

**Recommendation:** Accept with minor revision

**Comment:**

The manuscript has received generally positive evaluations, highlighting significant theoretical contributions and clear connections established between accelerated gradient methods and dynamical systems theory. The reviewers have confirmed that most of their concerns have been adequately addressed.

However, as pointed out by one reviewer, there remains an important aspect that requires additional clarification.
Comparison of ARR with standard quality metrics: The authors have partially addressed the requested comparison between the proposed Average Residual Ratio (ARR) metric and established quality metrics commonly used in evaluating t-SNE embeddings. Although the current version includes numerical experiments and visualization (Figures 9, 10, 11), it lacks a direct and explicit comparative discussion or quantitative analysis with standard metrics. For the final submission, the authors must explicitly address this concern by either providing a direct comparison or clearly articulating how ARR relates quantitatively and qualitatively to existing standard metrics. This will substantially enhance the practical relevance and interpretability of the ARR metric for practitioners.

**Audience:**

This paper offers novel theoretical insights into a widely-used dimensionality reduction method, t-SNE, especially when extended with accelerated gradient methods. The results are likely to be of interest to researchers and practitioners within the TMLR community, particularly those involved in dimensionality reduction, visualization, and optimization methods.

**Claims And Evidence:**

The claims made by the authors regarding the theoretical insights into the accelerated t-SNE dynamics, especially the connection to Bessel functions and closed-form solutions under momentum methods (MM and NAG), are supported convincingly by rigorous derivations and numerical evaluations. Reviewers unanimously acknowledge that the authors have provided accurate and clear evidence supporting their claims.